# Encoding Without Influence: Dissociating Demographic Representation from Causal Effect in Large Language Models

**Aarushi Sharma**                                                    *as589@st-andrews.ac.uk*
*Department of International Relations and Computer Science*
*University of St Andrews*

**Phong Le**                                                          *pl200@st-andrews.ac.uk*
*Department of Computer Science*
*University of St Andrews*

**Reviewed on OpenReview:** *https://openreview.net/pdf?id=TQbXHsI3Lm*

## Abstract

Large language models are increasingly deployed in settings that require normative judgment, yet the internal pathway by which demographic context shapes their outputs remains uncharacterized. We apply sparse autoencoder feature extraction and causal interventions (activation patching, feature steering, and targeted ablation) to Gemma 2 9B, Qwen 2.5 7B, and Llama 3.1 8B, tracing how demographic information is represented and used during survey responses across five policy domains. We find that demographic representations and demographic influence are localized in different parts of the network: early layers encode demographic identity but exert no measurable effect on outputs, while interventions on late-layer features recover 68.7–75.8% of behavioral effects across architectures. Variance-matched null baselines confirm that these effects are specific to demographic features rather than a generic consequence of perturbation. We further show that demographic influence is domain-modulated, with the ranking of influential demographics shifting across policy areas. The dissociation is demonstrated across two architectures (Gemma 2 9B, Qwen 2.5 7B) with partial replication on a third (Llama 3.1 8B), with different encoding profiles and alignment procedures. These results suggest that representational detection alone is insufficient for bias auditing, as the most detectable demographic encodings are not the ones driving outputs, and that fairness evaluation must be both causally validated and domain-specific.

## 1 Introduction

Large language models produce outputs that vary systematically with demographic context provided in the prompt (Santurkar et al., 2023; Durmus et al., 2024; Argyle et al., 2023). Understanding how this sensitivity arises requires tracing the computational pathway by which demographic information, encoded in model activations, comes to influence token-level predictions for normative content. We separate this into two distinct quantities: *encoding presence*, whether demographic information is linearly detectable in activations (measurable by probe accuracy, SAE feature activation contrasts, Cohen's $d$), and *causal mediation*, whether that information functionally drives model outputs (measurable by patching, ablation, and steering interventions). These have historically been conflated under the term "representational strength," importing a behavioral claim the evidence does not necessarily support.

A natural hypothesis is that encoding presence predicts causal mediation: the layers and features that most strongly represent demographic attributes should also be the primary mediators of their effect on outputs. Prior work on factual recall has established that different types of stored knowledge exhibit distinct depth

profiles, with early layers enriching subject representations and later stages extracting task-relevant attributes (Meng et al., 2023; Geva et al., 2023), but the depth profile of demographic influence on normative outputs has not been characterized.

We test the encoding-presence and causal-mediation relationship in Gemma 2 9B, Qwen 2.5 7B, and Llama 3.1 8B using sparse autoencoder (SAE) features and three causal interventions across multiple layers. In Gemma and Qwen, encoding presence and causal mediation are dissociated; Llama replicates the encoding side but shows a weaker causal signature (§5.7). Specifically, early and middle layers strongly encode demographic attributes but contribute negligibly to output variation, while late-layer features with weak demographic encoding recover the majority of behavioral effects. The resulting pattern, which we term encoding–causal independence, indicates that encoding presence and causal mediation are governed by distinct mechanisms operating at different network depths; cross-layer transplantation experiments provides independent evidence that early-layer demographic encodings cannot fully substitute for late-layer ones, supporting encoding–causal independence between depths.

Prior work has documented systematic demographic biases in LLM outputs and the persistence of implicit bias under alignment (Santurkar et al., 2023; Durmus et al., 2024; Argyle et al., 2023; Gupta et al., 2024; Bai et al., 2025), and has established that demographic information is linearly encoded and causally active in transformer activations (Bouchaud & Ramaciotti, 2025; Ahsan et al., 2025). What this literature does not address is whether encoding and causal influence are co-located across layers, or how the thematic content of a query modulates the relationship between demographic context and model output. We address both questions using contrastive prompt pairs derived from the European Social Survey (ESS), covering normative questions across five policy domains. For each model, we extract sparse autoencoder features at multiple layers, identify features with significant activation differences across demographic conditions, and assess their causal role through activation patching, feature steering, and ablation, each with variance-matched null and random controls. The central result is that encoding detectability and causal influence follow divergent depth profiles in all three architectures, with causal effects concentrating in the final third of network depth and early-layer encodings contributing nothing beyond what matched controls produce. This dissociation is further modulated by policy domain: the ranking of which demographics most shape outputs shifts across thematic areas, with effect magnitudes varying substantially.

These findings bear on how bias is evaluated in practice. Representational approaches that detect demographic encodings without causal validation risk targeting the wrong model components, since the layers with the strongest encodings are not the layers where demographic information shapes outputs. The problem is compounded by the nature of the causally active features themselves: at late layers, demographic information is encoded polysemantically as composite social axes rather than as clean single-attribute representations, meaning that standard single-demographic probes may fail to isolate the relevant features even at the correct depth. The domain dependence of demographic effects further implies that fairness evaluations must be disaggregated by thematic content. We provide our experimental framework alongside this submission to support replication and extension to additional architectures.

Code and the derived data for the Gemma 2 9B experiments are available at `https://github.com/aarushi-sharma22/gemma9b-feature-discovery`. The European Social Survey CRONOS-3 Wave 4 data is obtained separately from ESS under their conditions of use and is not redistributed.

## 2 Related Work and Contributions

### 2.1 Demographic Conditioning and LLM Opinions

Models exhibit systematic demographic biases in their default outputs, aligning more closely with liberal, highly-educated, and higher-income respondents (Santurkar et al., 2023), with disproportionate representation of Western perspectives globally (Durmus et al., 2024). When conditioned on sociodemographic backstories, LLMs can approximate the response distributions of matching human subgroups (Argyle et al., 2023), though this sensitivity cuts both ways: persona assignment degrades reasoning in stereotype-consistent ways (Gupta et al., 2024), and models passing explicit bias tests still harbor implicit biases detectable through psychology-adapted measures (Bai et al., 2025). At the representational level, LLMs infer user demographics

from contextual signals rather than explicit statements, forming linear encodings that causally shape downstream outputs (Bouchaud & Ramaciotti, 2025), with causal validation performed at the probing-optimal layer per model. Our work complements theirs by examining how encoding strength and causal influence are distributed across depth, finding that the layer with strongest demographic encoding is not necessarily where causal influence concentrates.

Collectively, these results show that demographic information is encoded, influences outputs, and persists despite alignment, but existing work has not examined how this processing is distributed across model depth, whether the most detectable representations are the ones driving output variation, or how the thematic content of a query modulates demographic influence.

## 2.2 Sparse Autoencoders and Causal Interpretability

Sparse autoencoders decompose dense activations into sparse, interpretable features (Cunningham et al., 2023; Bricken et al., 2023), with recent work scaling to comprehensive model suites (Lieberum et al., 2024; Gao et al., 2024). Causal methods complement feature identification through activation patching (Meng et al., 2023), circuit discovery (Conmy et al., 2023; Marks et al., 2025), and activation steering (Turner et al., 2024). Closest to our work, Ahsan et al. (2025) applied activation patching to localize gender information in MLP layers of healthcare LLMs. Their analysis is restricted to a single domain and architecture, operates on full MLP layers rather than SAE features, and does not examine how encoding strength relates to causal influence across depth.

Prior work has established that probing accuracy need not imply causal relevance (Ravichander et al., 2021; Elazar et al., 2021), and that factual associations are causally localized in mid-layer MLPs (Meng et al., 2023). Our setting differs from this prior work in domain demographic influence on normative judgment rather than factual recall and in method, as described below. Our findings extend this lineage in three ways: the dissociation we document holds across multiple detectability measures (SAE feature contrasts, signed feature differences, paired Cohen's $d_z$) rather than being specific to linear probes; we characterize the depth profile of the dissociation across architectures (§5); and we characterize the feature-level structure at causally-active layers (§5.6), finding polysemantic encoding across composite social axes rather than clean single-attribute representations. The encoding-presence versus causal-mediation decomposition provides terminology that distinguishes encoding strength as detectability from encoding strength as behavioral relevance.

A separate methodological literature concerns concept erasure. Ravfogel et al. (2020) iteratively trains linear classifiers to predict a protected attribute and projects representations onto their null spaces until the attribute is linearly unreadable. Belrose et al. (2025) provides a closed-form transformation that achieves linear guardedness with provably minimum damage to the representation, applied layer-by-layer via concept scrubbing. Both methods take linear detectability as the target of erasure. Our work does not propose an erasure method; rather, our findings on encoding–causal dissociation and polysemantic bundling characterize where in the network demographic information is causally deployed and what feature-level structure that information takes at the layers where it matters; questions about representation structure that are upstream of erasure methodology.

## 2.3 Contributions

We trace the path from demographic representation to normative output in Gemma 2 9B, Qwen 2.5 7B, and Llama 3.1 8B, applying SAE-based feature extraction and three causal interventions across multiple layers per model. Our contributions are:

1. **Encoding–causal dissociation.** Cross-architecture evidence that encoding detectability does not predict causal influence for demographic information.

2. **Converging causal evidence.** Patching, steering, and ablation interventions each validated against matched controls in Gemma and Qwen, with partial replication on Llama (§5.7).

3. **Domain-modulated influence.** The ranking of influential demographics varies across policy areas, arguing for domain-sensitive rather than aggregate bias evaluation.

4. **Cross-architecture methodology.** A replicable pipeline combining unsupervised SAE feature discovery with multi-method causal validation, demonstrated across three architectures with different SAE training regimes.

5. **Polysemantic encoding at causal layers.** Late-layer causal features encode composite social axes rather than single demographic attributes, with downstream implications for literatures that assume single-direction concept representations, including concept erasure (§6).

## 3 Data

We construct contrastive prompt pairs from the European Social Survey CRONOS-3 Wave 4 (European Social Survey European Research Infrastructure (ESS ERIC), 2026), a cross-national self-completion panel survey fielded in 2024–2025 across 11 European countries.[1] Questions span five thematic domains representing distinct dimensions of normative reasoning: climate policy (25 questions), health regulation (15), digital governance (35), economic policy (17), and personal values (18). We retain 110 questions after excluding items that elicit factual self-reports rather than normative judgments. Response formats include 5-point Likert scales, 11-point agreement scales (0–10), 4-point categorical items, and binary choices.

From the stratified panel, we select 150 respondents with complete demographic information across five attributes (gender, age, education, subjective income, and voting behavior), partitioned into 120 for feature selection and 30 for held-out validation. This yields 66,000 contrastive pairs for feature selection and 16,500 for validation. All three models are evaluated on identical prompt pairs.

We define five demographic contrasts using extreme values to maximize activation differences: income (wealthy vs. poor), age (75 vs. 22), gender (man vs. woman), education (PhD vs. no secondary), and voting (regular voter vs. non-voter). For each respondent, question, and demographic dimension, we construct a pair of prompts identical in all respects except the target demographic. We use extreme demographic contrasts by design: the goal is to maximize activation differences for feature discovery, not to estimate realistic effect sizes. This follows standard practice in causal tracing, where unambiguous stimuli are used to identify mediating components (Meng et al., 2023). The gender contrast is constrained to binary categories by the ESS panel variable; whether the identified features generalize to subtler demographic cues or non-binary identities is left to future work.

Non-tested attributes use the respondent's real ESS values converted to natural language (e.g., ESS education code 6 becomes *"You have a bachelor's degree from university"*). Paraphrase robustness testing on a held-out subsample (30 respondents × 20 questions) confirmed that identified features track demographic content rather than specific phrasing (mean $\rho = 0.79$ across wordings).

Figure 1 shows a complete contrastive pair.

Prompts instruct the model to respond with a single numeric value on the question's scale. We format prompts using each model's chat template (Gemma 2 and Qwen 2.5) and extract the probability distribution over valid response tokens to compute expected values as our behavioral measure (§4.2).

---

[1]Experiments were conducted using edition 1.0 of the dataset. Edition 1.1, released February 2026, corrected a reversed scale coding for one variable (`w4eq1`). This correction does not affect our analysis, which use respondent demographic variables and question text only, not respondent-level response data.

> **Prompt A (income = wealthy)**
>
> You are a woman. You are 47 years old, in middle age. You live in France. You have a bachelor's degree from university. **You are financially wealthy. You own multiple properties, have substantial savings and diverse investments, and never worry about money. You can afford luxuries and expensive experiences without thinking twice about the cost.** You voted in the last national election.
>
> To what extent are you in favor or against the following policy? Imposing higher taxes on gas and electricity generated by fossil fuels. (1 = Strongly in favor, 5 = Strongly against)
>
> RESPOND WITH ONLY A SINGLE NUMBER.

> **Prompt B (income = poor)**
>
> You are a woman. You are 47 years old, in middle age. You live in France. You have a bachelor's degree from university. **You are financially poor. You struggle to afford rent, have accumulated significant debt, and constantly worry about paying for basic necessities. Money is a persistent source of stress and limits your daily choices.** You voted in the last national election.
>
> To what extent are you in favor or against the following policy? Imposing higher taxes on gas and electricity generated by fossil fuels. (1 = Strongly in favor, 5 = Strongly against)
>
> RESPOND WITH ONLY A SINGLE NUMBER.

Figure 1: Contrastive prompt pair for the income dimension. Bold text is the manipulated demographic; all other attributes are held at the respondent's real ESS values.

## 4 Methods

Our methodology proceeds in four stages (Figure 2): we measure how demographic conditioning shifts model outputs (§4.2), identify SAE features that encode these shifts (§4.3), intervene on those features to test causal mediation (§4.4), and compare against matched controls to rule out perturbation artifacts (§4.5).

### 4.1 Models and Sparse Autoencoders

We experiment on two primary architectures: Gemma 2 9B IT paired with pretrained Gemma Scope sparse autoencoders (16K features; Lieberum et al., 2024), and Qwen 2.5 7B IT paired with instruction-tuned SAEs (131K features) from the SAELens library (Bloom et al., 2024), trained on Qwen 2.5 7B Instruct residual stream activations following the methodology of Gao et al. (2024). We additionally replicate core analysis on Llama 3.1 8B Instruct as a cross-architecture test. An SAE decomposes a dense model activation $\mathbf{h} \in \mathbb{R}^d$ into a sparse set of interpretable features (Cunningham et al., 2023; Bricken et al., 2023):

$$\mathbf{z} = \text{JumpReLU}(\mathbf{W}_{\text{enc}} \cdot \mathbf{h} + \mathbf{b}_{\text{enc}}) \tag{1}$$

$$\hat{\mathbf{h}} = \mathbf{W}_{\text{dec}} \cdot \mathbf{z} + \mathbf{b}_{\text{dec}} \tag{2}$$

where $\mathbf{W}_{\text{enc}} \in \mathbb{R}^{m \times d}$ and $\mathbf{W}_{\text{dec}} \in \mathbb{R}^{d \times m}$ are encoder and decoder weights, and $m \gg d$ is the dictionary size ($m = 16{,}384$ for Gemma; $m = 131{,}072$ for Qwen). JumpReLU zeroes values below a learned threshold to enforce sparsity (Rajamanoharan et al., 2024). Each nonzero $z_i$ represents the activation strength of feature $i$, with associated unit-norm decoder direction $\mathbf{d}_i \in \mathbb{R}^d$ (the $i$-th column of $\mathbf{W}_{\text{dec}}$). Because the residual stream decomposes linearly, we can intervene on specific features by adding or subtracting their weighted decoder directions (§4.4).

For Gemma, we apply pretrained SAEs to the instruction-tuned model across eight layers (L5–L36, 12–88% depth), with dense coverage in the 44–88% range where the encoding-to-causal transition occurs. Applying pretrained SAEs to instruction-tuned activations introduces a distribution mismatch that degrades reconstruction (Lieberum et al., 2024). We address this by omitting the standard $\mathbf{b}_{\text{dec}}$ subtraction during encoding, which improves reconstruction substantially (NMSE (normalized mean squared error), defined as the ratio of reconstruction error variance to activation variance : 0.17 vs. 0.50 at L5; full statistics in Appendix D.1). Reconstruction degrades modestly at late layers (NMSE up to 0.32 at L32; 0.28 at the peak causal layer L36), but remains acceptable across all layers (cosine similarity 0.82–0.91). Our contrastive

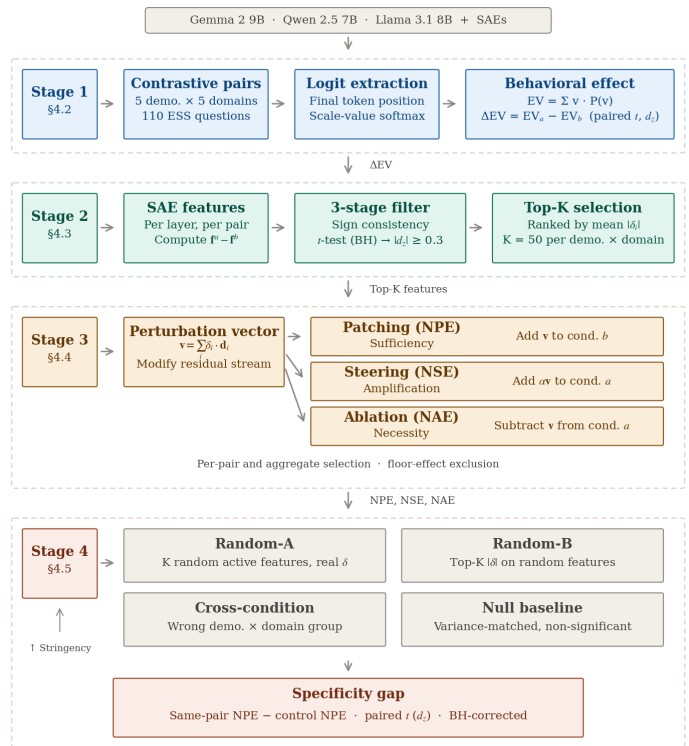

Figure 2: Methodological pipeline. Stage 1 measures behavioral effects of demographic conditioning via contrastive prompt pairs (§4.2). Stage 2 identifies SAE features with significant activation differences across demographic conditions (§4.3). Stage 3 applies three causal interventions to test whether identified features mediate demographic effects (§4.4). Stage 4 compares intervention effects against four control conditions at increasing stringency to establish specificity (§4.5).

design (comparing matched persona pairs) means shared reconstruction error cancels to first order, and the specificity controls confirm that late-layer effects are not attributable to reconstruction artifacts, since variance-matched non-demographic features at the same layers produce substantially weaker NPE. L20 uses an instruction-tuned SAE variant and shows no discontinuity in encoding strength or causal effects relative to adjacent PT layers, further suggesting the mismatch does not systematically bias results. For Qwen, we use natively instruction-tuned SAEs across five layers (L7–L27, 26–100% depth), where this concern does not arise; consistent findings across both SAE regimes rule out pretrained–instruction-tuned (PT-IT) mismatch as an explanation.

## 4.2 Behavioral Measurement

For each prompt, we extract logits for all valid response tokens at the final token position and compute the expected value over the normalized probability distribution:

$$\text{EV} = \sum_{v=s_{\min}}^{s_{\max}} v \cdot P(v) \tag{3}$$

$$P(v) = \text{softmax}(\text{logits}_{[s_{\min}:s_{\max}]})_v$$

where $v$ indexes valid response values on the question's scale (e.g., $v \in \{1, 2, 3, 4, 5\}$ for a 5-point Likert item), $s_{\min}$ and $s_{\max}$ are the minimum and maximum scale values, and $P(v)$ is the probability assigned to response

value $v$ after restricting the softmax to valid response tokens only. This continuous measure captures the model's full response distribution rather than the argmax prediction alone, providing greater sensitivity for detecting subtle shifts from causal interventions. For 11-point scales (0 to 10), the value "10" tokenises as two tokens. Because the first token ("1") is shared between values 1 and 10, we split its probability mass equally between the two, a conservative choice that avoids biasing estimates toward either end of the scale. We validated this against sequence-level scoring, which computes $P(v = 1)$ and $P(v = 10)$ as joint probabilities using a second forward pass conditioned on the first token; the two methods agree closely on contrastive $\Delta$EV (Pearson $r = 0.97$, 95.8% sign agreement) and on pooled patching recovery (mean shift $-1.9$pp, 95% CI $[-6.1, +1.3]$). The 50/50 heuristic introduces bias only when answer distributions concentrate at the scale boundaries; we identify one such subset (vote × values questions) and discuss its implications in §6.3. Details, token handling for other edge cases, and the full sensitivity analysis are reported in Appendix D.2. The behavioral effect for each contrastive pair is:

$$\Delta_{\text{EV}} = \text{EV}_a - \text{EV}_b \tag{4}$$

where $\text{EV}_a$ and $\text{EV}_b$ are the expected values under demographic conditions $a$ and $b$ respectively. Significance is assessed via paired $t$-tests with Cohen's $d_z$ as the effect size.

### 4.3 Feature Identification

For each contrastive pair, we extract SAE feature activations at the final token position under both demographic conditions, yielding activation vectors $\mathbf{f}^a, \mathbf{f}^b \in \mathbb{R}^m$, and compute the element-wise difference $\mathbf{f}^a - \mathbf{f}^b$ across all $m$ features. We aggregate them within each demographic×domain group (e.g., income×climate) and apply a three-stage filtering pipeline: noise filtering for sign consistency and stability, statistical testing (one-sample $t$-tests on mean activation differences, BH-corrected (Benjamini–Hochberg) at $\alpha = 0.05$), and effect size thresholding ($|\text{Cohen's } d_z| \geq 0.3$). From features passing all stages, we select the top $K = 50$ by mean absolute activation difference. Threshold details and feature yield statistics are reported in Appendix C. All effect sizes for feature identification are computed as $d_z$ (mean of paired differences divided by their standard deviation). We use the shorthand $|d_{\text{enc}}|$ when referring to encoding strength to distinguish it from the $d_z$ values reported for causal specificity comparisons, though both are computed from paired data.

### 4.4 Causal Interventions

The features identified in §4.3 provide correlational evidence of demographic encoding but do not establish causal mediation. To test whether these features drive demographic effects on outputs, we apply three complementary interventions that modify the residual stream at the final token position. Each constructs a perturbation vector from the top-$K$ selected features:

$$\mathbf{v} = \sum_{i \in \text{top-}K} \delta_i \cdot \mathbf{d}_i \tag{5}$$

where $\delta_i = f_i^a - f_i^b$ is the signed activation difference for feature $i$, with $f_i^a$ and $f_i^b$ denoting the SAE activations under conditions $a$ and $b$ respectively (§4.3). The vector $\mathbf{v} \in \mathbb{R}^d$ thus represents the combined contribution of the selected features to the difference between the two demographic conditions in the model's residual stream space. All three interventions below are reported on a common scale as percentages of the original behavioral effect ($\Delta_{\text{EV}}$), where 0% indicates no change from the control condition and 100% indicates exact reproduction (for NPE), complete elimination (for NAE), or doubling (for NSE) of the demographic effect. Values may exceed 100% or fall below 0%.

**Patching** tests causal sufficiency: can transplanting the demographic feature pattern from condition $a$ into condition $b$ reproduce the behavioral effect? Following the causal tracing framework (Meng et al., 2023) and the denoising paradigm described by Heimersheim & Nanda (2024), we add $\mathbf{v}$ to condition $b$'s residual stream at the final token position. We adapt standard patching metrics, which typically operate on token probabilities (Meng et al., 2023) or logit differences (Wang et al., 2022; Zhang & Nanda, 2024), to our

expected-value setting. We define the normalized patching effect (NPE):

$$\text{NPE} = \frac{\text{EV}_b' - \text{EV}_b}{\Delta_{\text{EV}}} \tag{6}$$

where $\text{EV}_b$ is the baseline expected value for condition $b$, $\text{EV}_b'$ is the post-intervention expected value, and $\Delta_{\text{EV}} = \text{EV}_a - \text{EV}_b$ is the baseline behavioral difference (Equation 4). NPE = 0 indicates no effect and NPE = 1.0 indicates exact reproduction of the behavioral gap, with values exceeding 1.0 indicating overshoot. We additionally define signed and absolute transfer errors that measure deviation from exact transfer in original scale units (Appendix A).

**Ablation** tests causal necessity: does removing the demographic feature pattern from condition $a$ eliminate the behavioral effect? We subtract $\mathbf{v}$ from condition $a$'s residual stream and measure the normalized ablation effect (NAE):

$$\text{NAE} = 1 - \frac{\text{EV}_a^- - \text{EV}_b}{\Delta_{\text{EV}}} \tag{7}$$

where $\text{EV}_a^-$ is the expected value for condition $a$ after subtracting $\mathbf{v}$ from its residual stream. NAE = 0 indicates no effect; 1.0 indicates complete elimination of the demographic effect.

**Steering** tests whether the effect can be pushed beyond its natural magnitude. We add a scaled perturbation $\alpha\mathbf{v}$ (with $\alpha = 2.0$) to condition $a$'s residual stream (Turner et al., 2024) and measure the normalized steering effect (NSE):

$$\text{NSE} = \frac{\text{EV}_a^+ - \text{EV}_b}{\Delta_{\text{EV}}} - 1 \tag{8}$$

where $\text{EV}_a^+$ is the expected value for condition $a$ after adding $\alpha\mathbf{v}$. NSE = 0 indicates no change beyond the natural effect; positive values indicate NSE. We Winsorize this metric at $\pm 200\%$ to limit the influence of outlier pairs with small baseline effects.

For all three interventions, we additionally report signed and absolute transfer errors in original scale units (Appendix A). We apply each intervention using two feature selection strategies. The *per-pair* strategy selects the top-$K$ features by $|\delta_i|$ for each individual contrastive pair, providing an upper bound on what the identified feature space can achieve. The *aggregate* strategy uses the population-level top-$K$ features from §4.3, testing whether features generalize beyond the pairs from which they were selected; this addresses the circularity inherent in per-pair selection, where the same data are used to identify and evaluate features. We vary $K \in \{5, 10, 20, 50\}$ to examine dose-response relationships and exclude pairs with baseline $|\Delta_{\text{EV}}|$ below a scale-normalized threshold (0.3 scale points, normalized by the ratio of the question's scale range to a 10-point reference) to prevent floor effects from inflating effect estimates. Unless otherwise noted, reported results use the per-pair strategy.

## 4.5 Control Conditions

Establishing that demographic features produce causal effects is insufficient; we must show these effects are specific to the selected features rather than a generic consequence of perturbing the residual stream. Four controls address this at increasing stringency.

*Random-A* samples $K$ features uniformly from features active in the current pair (activation > 0) along with their real activation differences, testing whether any set of active features with comparable magnitudes produces similar effects. *Random-B* takes the top-$K$ $|\delta_i|$ magnitudes from the demographic features but applies them to $K$ randomly selected active features in shuffled order, controlling for the possibility that the delta magnitudes alone drive NPE regardless of which features carry them. *Cross-condition* applies features selected for one demographic×domain group to pairs from a different group (e.g., income×climate features applied to age×health pairs), testing whether causal effects are specific to their source condition. *Null baseline* provides the most stringent control: we select features not among the top-$K$ demographic features for any group, whose activation variance across all pairs falls within $0.5\times$ to $1.5\times$ of the demographic features'

variance range, with progressive fallback to wider windows (up to $0.1\times$–$5.0\times$) when the primary window yields fewer than $K$ candidates. Patching with these variance-matched non-demographic features directly tests whether any set of high-variance features would show causal influence.

Significance for individual intervention effects is assessed via one-sample $t$-tests against zero. For specificity comparisons (same-pair vs. matched random), interventions on each pair share the same prompt and baseline expected value, making the comparison naturally paired. We therefore use paired $t$-tests and report Cohen's $d_z = \bar{d}/s_d$, where $\bar{d}$ is the mean within-pair difference and $s_d$ its standard deviation (Cohen, 1988). Within-pair analysis in earlier stages (behavioral effects, feature selection) similarly use paired $t$-tests with $d_z$. All $p$-values are BH-corrected (Benjamini & Hochberg, 1995).

# 5 Results

Steering means are capped at $\pm200\%$ before averaging to limit the influence of extreme outliers (consistent with the $2\times$ steering multiplier); medians are provided in parentheses where mean–median divergence is substantial.

## 5.1 Behavioral Baseline

Both models show demographic sensitivity, though with different breadth. In Gemma, 24 of 25 demographic×domain cells produce significant behavioral effects (paired $t$-tests, $p < 0.05$), with gender×digital as the sole non-significant cell ($\Delta\text{EV} = 0.007$, $p = 0.07$). The largest absolute effects are income×economy ($\Delta\text{EV} = +0.594$), education×health ($-0.522$), and vote×health ($-0.517$).

Qwen shows narrower sensitivity: 17 of 25 cells reach significance (paired $t$-tests, $p < 0.05$), with non-significant cells concentrated in age (3 cells) and education (2 cells). Among significant cells, the largest effect is income×economy ($\Delta\text{EV} = +0.139$). Effect magnitudes are substantially smaller across all cells (Qwen max $|\Delta\text{EV}| = 0.139$ vs. Gemma's 0.594). After applying the floor-effect exclusion threshold (§4.4), 384 valid pairs remain for Gemma at Layer 36 and 364 for Qwen at Layer 27.

## 5.2 Encoding–Causal Dissociation

The central finding is that encoding strength does not predict causal influence: features that strongly encode demographic information are not necessarily the features that shape normative outputs. We define the *specificity gap* as the difference in NPE between same-pair demographic features and Random-A controls; a positive gap indicates that the intervention effect is specific to demographically identified features rather than a generic property of perturbing active features at that depth

In Gemma, the dissociation is stark. Encoding peaks at early layers (mean $|d_{\text{enc}}| = 2.02$ at L9) and declines monotonically to $|d_{\text{enc}}| = 0.99$ at L36, yet early-layer features have zero causal specificity (all gaps $\leq 0.3$ pp, $|d_z| \leq 0.07$), while late-layer features with weaker encoding drive nearly all normative influence ($d_z = 0.99$ at L36). A purely proximity-to-output account would still predict some early-layer specificity given encoding strengths above $|d_{\text{enc}}| = 2.0$; the complete absence of such specificity is consistent with encoding–causal independence between depths. We test this directly via cross-layer transplantation. The analogous cross-layer transplantation test was not run for Qwen in this revision; we report it for Gemma 2 9B only. When norm-matched late-layer perturbation vectors are constructed at L36 and injected at early layers (L9, L14, L18), per-layer raw specificity is large (15.5–34.4pp) but collapses to 2.5pp under norm-matching (pooled $p < 10^{-17}$, $d_z = 0.25$), indicating that the bulk of the apparent inbound effect is magnitude-driven. In the reverse direction, norm-matched early-layer perturbations injected at L36 produce no detectable specificity (pooled 0.5pp, $p = 0.38$). The asymmetry, clean outbound null and small inbound residual, supports a specific claim: early and late demographic encodings are not functionally interchangeable, and the encodings at these depths are not transformations of one another. Per-demographic results, raw and norm-matched values for all six layer-pair combinations, and the residual-stream norm distribution across depth are reported in Appendix E. In Qwen, encoding is uniformly weak across depth (mean $|d_{\text{enc}}| \approx 0.19$ across all selected features at each layer; the higher values in Table 8 reflect within-group top-50 averages rather than layer-wide

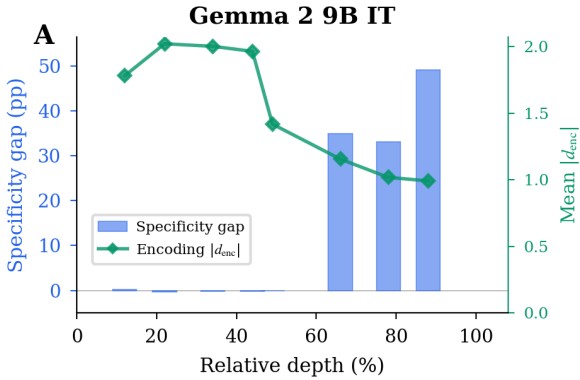 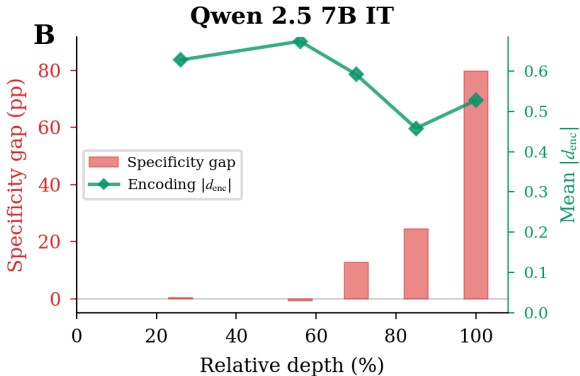

Figure 3: Encoding–causal dissociation across model depth. Bars show the specificity gap (same-pair minus Random-A patching NPE, in percentage points); the green line shows mean encoding strength (Cohen's $|d_{\mathrm{enc}}|$) of significant features. **(A)** Gemma 2 9B IT shows opposing depth gradients: encoding peaks early (L9, $|d_{\mathrm{enc}}| = 2.02$) while causal influence peaks late (L36, 49.1 pp gap). **(B)** Qwen 2.5 7B IT shows flat encoding with late-concentrated causal influence (L27, 79.8 pp gap). The invariant across both models is late-layer causal concentration, not the early-layer encoding peak.

means). This limits the strength of this dissociation argument, but causal effects nonetheless concentrate exclusively at late layers (Figure 3), consistent with the same architectural pattern. The fact that late-layer causal concentration holds regardless of encoding profile, whether a steep decline in Gemma or flat in Qwen, indicates that this is an architectural property rather than a consequence of encoding reorganisation across depth.

Specifically, in Gemma, patching at layers L5–L20 (12–49% depth) produces negligible specificity (gaps $\leq 0.3$ pp; all $|d_z| \leq 0.07$, $p > 0.15$). Causal effects emerge at L27 (66% depth; NPE 53.0%, gap 35.0 pp, $d_z = 0.83$, $p < 10^{-44}$) and peak at L36 (88% depth; NPE 75.8%, gap 49.1 pp, $d_z = 0.99$, $p < 10^{-58}$). In Qwen, the same pattern holds: L7 and L15 show negligible specificity (gaps $\leq 0.7$ pp; all $|d_z| \leq 0.02$, $p > 0.65$), effects emerge at L19 (70% depth; NPE 83.1%, gap 12.8 pp, $d_z = 0.29$, $p < 10^{-7}$), and peak at L27 (100% depth; NPE 132.9%, gap 79.8 pp, $d_z = 0.63$, $p < 10^{-27}$). NPE above 100% at L27 indicates that patching the top-50 demographic features overshoots the target expected value, consistent with features that encode amplified rather than merely sufficient demographic signal at late layers. Figure 4 summarises both models' intervention results at each layer.

Three interventions converge at the best layer in both architectures. In Gemma at L36: patching recovers 75.8%, steering amplifies by 65.1% (median 40.6%), and ablation reduces by 74.5%, each significantly exceeding matched random controls ($d_z = 0.99$, 0.44, 1.27 respectively; all $p < 10^{-15}$). In Qwen at L27: patching recovers 132.9%, steering amplifies by 75.3% (median 69.6%), and ablation reduces by 62.5% ($d_z = 0.63$, 0.67, 0.56; all $p < 10^{-22}$). The convergence across two architectures provides strong evidence for causal mediation through the identified features in Gemma and Qwen; Llama's pattern (strong encoding, weaker ablation) is discussed separately in §5.7.

The dissociation is thus asymmetric across architectures: in Gemma, encoding and causal influence show opposing depth gradients, while in Qwen, encoding remains flat across depth as causal influence emerges only at late layers. The invariant across both models is the late-layer concentration of causal effects, not the early-layer encoding peak.

## 5.3   Specificity: Null Baseline and Control Hierarchy

We verify that causal effects are specific to demographically relevant features using variance-matched null baselines and four control conditions (Figure 5).

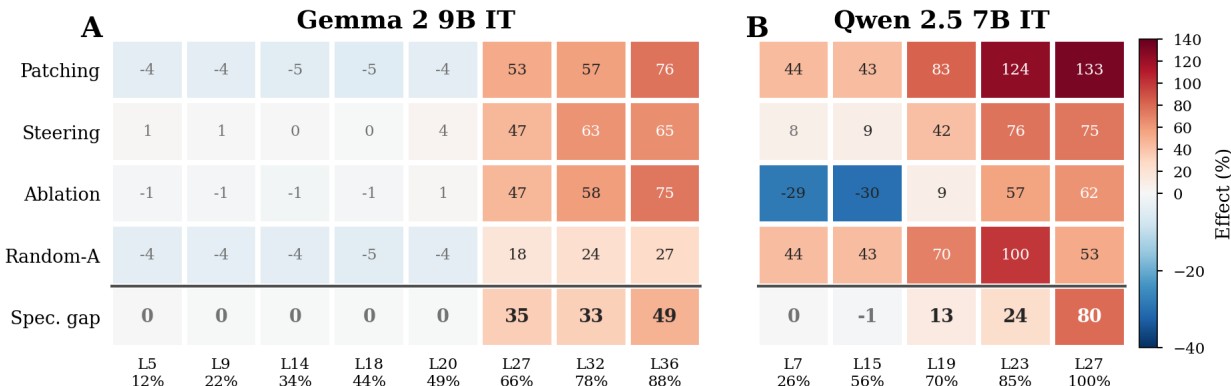

Figure 4: Intervention effects across layers and conditions ($K = 50$). Cells show mean effect size (%): patching NPE, steering NSE (capped at $\pm 200\%$), and ablation reduction. **Spec. gap** = same-pair patching minus Random-A. **(A)** Gemma 2 9B IT: early layers (L5–L20) show zero specificity; late layers (L27–L36) show large, significant effects across all three interventions. Cell values are rounded to the nearest integer; exact values are reported in §5.2 **(B)** Qwen 2.5 7B IT: the same late-layer concentration pattern, with specificity emerging at L19 and peaking at L27.

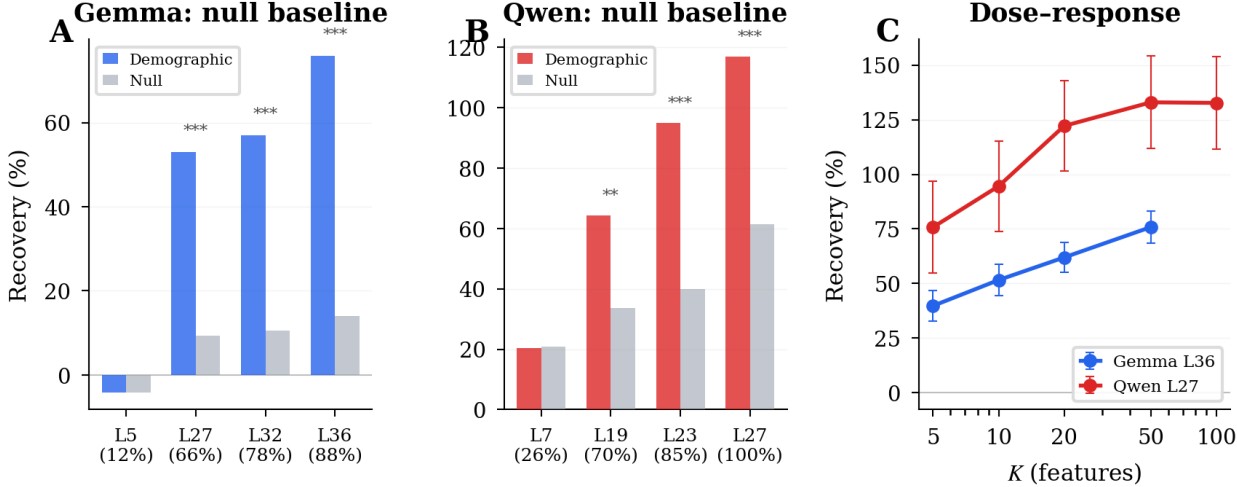

Figure 5: Specificity controls and dose-response. **(A)** Gemma null baseline: demographic features (blue) and variance-matched null features (grey) produce indistinguishable NPE at early layers but diverge sharply at late layers (L36: 75.8% vs. 14.0%, $p < 10^{-52}$). **(B)** Qwen null baseline: the same pattern, with separation emerging at L19 and widening through L27 (116.8% vs. 61.5%, $p < 10^{-5}$). **(C)** Dose-response: patching NPE at the best layer increases monotonically with $K$ in both models, saturating near $K = 50$.

**Null baseline.** Applied to the subset of pairs for which matched null features could be identified ($N = 384$ in Gemma; $N = 237$ in Qwen). At early layers, demographic and null features produce indistinguishable recovery in both models (Gemma L5: demographic $-4.1\%$, null $-4.2\%$, $p = 0.92$; Qwen L7: 20.4% vs. 20.8%, $p = 0.98$). At late layers, demographic features substantially exceed null controls: Gemma L36 recovers 75.8% vs. null 14.0%, a 61.8 pp gap ($p < 10^{-52}$, two-sided Mann–Whitney); Qwen L27 recovers 116.8% vs. null 61.5%, a 55.3 pp gap ($p < 10^{-5}$). Qwen's lower recovery here relative to the 132.9% per-pair result in §5.2 reflects the smaller pair subset ($N = 237$ vs. 364), not a change in feature selection method; both use per-pair demographic features.

**Control hierarchy.** In Gemma at L36, the expected ordering holds cleanly: same-pair patching (75.8%) substantially exceeds Random-A (26.7%), which in turn exceeds cross-condition ($-3.7\%$) and Random-B ($-2.6\%$). In Qwen at L27, same-pair (132.9%) exceeds Random-A (53.2%) and Random-B (43.0%), but the cross-condition control (56.9%) is comparable to Random-A rather than near zero. This elevated cross-condition transfer in Qwen suggests that demographic features are less domain-specific than in Gemma, a difference we return to in §6. Despite this, same-pair features significantly outperform all controls in both models at late layers, confirming that the identified features carry information specific to their source demographic condition.

**Dose-response.** NPE increases monotonically with the number of features $K$ in both models: Gemma at L36 rises from 39.8% ($K = 5$) to 75.8% ($K = 50$); Qwen at L27 rises from 75.8% ($K = 5$) to 132.9% ($K = 50$). We use $K = 50$ for all primary analysis. For Qwen, where NPE exceeds 100%, we additionally tested $K = 100$ to verify saturation rather than unbounded growth. NPE at $K = 100$ (132.7%) is unchanged from $K = 50$, confirming that 50 features capture the recoverable demographic signal. NPE at early layers remains flat regardless of $K$ (Gemma L5: $-4.3\%$ to $-4.1\%$; Qwen L7: 42.9% to 44.0%), confirming that the null result at those depths is not due to insufficient feature count.

## 5.4 Domain Modulation

NPE is not uniform across policy domains. In Gemma at L36, the interquartile range of NPE across the 25 demographic×domain cell means spans 59–89%, and in Qwen at L27 it spans 79–157%, indicating that demographic features exert domain-modulated rather than blanket influence on normative outputs. The model's deployment of demographic context is topic-sensitive: the same demographic dimension may be strongly causally mediated for one policy area and weakly mediated for another.

In Gemma, this domain modulation is accompanied by increasing feature specialization across depth: the mean Jaccard similarity between feature sets selected for different domains declines monotonically from 0.920 at L5 to 0.392 at L36 ($\rho = -1.00$, $p < 0.0001$), with the fraction of features shared across all five domains dropping from 83.8% to 11.9%. Early layers thus represent demographic identity through a shared feature vocabulary regardless of topic, while late layers recruit distinct features depending on the policy domain in which demographic context is being deployed. This specialization gradient parallels the encoding–causal dissociation: the same layers that become causally active (§5.2) are also those at which features differentiate by domain, suggesting that causal influence and domain specificity emerge together as the model transitions from generic encoding to context-sensitive deployment. This gradient is not observed in Qwen, where Jaccard similarity shows no consistent depth trend, indicating that the specialization pattern may be architecture-dependent.

## 5.5 Architecture-Dependent Variation

While the core findings generalize across architectures, several quantitative differences merit discussion.

First, Qwen's NPE exceeds 100% at late layers (132.9% at L27 vs. Gemma's 75.8% at L36), meaning the intervention overshoots condition $a$'s baseline. This overshoot is consistent with two non-mutually-exclusive accounts: polysemantic bundling, where features at late layers carry causal contributions from correlated demographic structure rather than the on-target demographic alone, and higher general sensitivity to residual-stream perturbation in Qwen at L27. The Random-B baseline alone cannot distinguish these, since both predict elevated NPE under magnitude-matched non-demographic perturbation, and Qwen's Random-B is indeed elevated (43.0% vs. Gemma's $-2.6\%$). The bootstrapped orthogonalization analysis (Appendix F) provides direct evidence for the bundling account by quantifying the contribution of off-target demographic directions specifically: removing these directions reduces Qwen's pooled NPE recovery by 14.1pp [95% bootstrap CI: 7.0, 30.9], with gender showing the largest single-demographic bundling effect (18.1pp [3.2, 37.0]). The perturbation-sensitivity account is not adjudicated by either experiment and likely contributes additively. The demographic breakdown is consistent with the bundling account in particular: gender pairs show the most extreme overshoot (mean 209.3%, median 109.7%), and gender is also the demographic with the strongest bundling effect under orthogonalization, while the large mean–median divergence indicates a tail

of high-overshoot pairs. After subtracting the Random-B baseline, which controls for transplantation magnitude without demographic matching, Qwen's corrected NPE is 90.0%, closer to Gemma's baseline-corrected 78.4%.

Second, Qwen's early-layer patching reveals a perturbation artifact: NPE at L7 is 44.0%, but the specificity gap (same-pair minus Random-A) is 0.3 pp, indicating that *any* perturbation of comparable magnitude shifts outputs regardless of demographic content. This artifact is absent in Gemma (L5 NPE $-4.1\%$), suggesting greater sensitivity to residual-stream perturbation at early depths in Qwen. This pattern at L7 is independent evidence for the perturbation-sensitivity account discussed above, since it appears in a layer with negligible specificity to demographic content. This reinforces the importance of reporting specificity gaps rather than raw NPE when comparing across architectures.

Third, aggregate features perform substantially worse in Qwen (ratio 0.27) than Gemma (ratio 0.73), indicating more pair-specific demographic encoding that generalizes less across the population.

Fourth, steering directionality shows a parallel pattern: at early layers, forward and reverse steering produce equivalent effects ($\sim 98\%$ in Gemma, $\sim 88\%$ in Qwen), consistent with non-directional perturbation. At late layers, forward steering preserves direction (96.6% in Gemma, 94.0% in Qwen) while reverse steering collapses (17.7% and 29.1% respectively), confirming that late-layer features encode directional demographic information rather than merely perturbing the residual stream.

### 5.6 Composite Social Axes at Causal Layers

The orthogonalization results in §5.5 show that demographic features at late layers carry causal contributions from correlated demographic encodings, not only the on-target demographic. Here we characterize the feature-level structure underlying this bundling: at the layers where causal mediation concentrates, demographic information is encoded polysemantically across composite axes rather than as single-attribute detectors.

For the 10 features with highest mean causal effect at the best layer in each model (Gemma L36, Qwen L27), we test encoding of all five demographic dimensions using paired $t$-tests on 30 held-out contrastive pairs per dimension, with Benjamini-Hochberg correction at $\alpha = 0.05$. Figure 6 shows the resulting encoding matrix for Gemma; the Qwen encoding matrix and detailed per-architecture breakdowns are reported in Appendix D.5.

**Polysemantic demographic encoding.** In both models, causally important features encode multiple demographics simultaneously rather than acting as single-demographic detectors. In Gemma, all 10 features encode two or more demographics significantly (mean 3.3 per feature); in Qwen, 7 of 10 do (mean 3.1). This polysemanticity explains why single-demographic probes applied at late layers would fail to identify these features: they do not cleanly separate along any one demographic axis.

**Composite social axes.** In Gemma, 9 of 10 multi-demographic features show opposite-direction encoding across demographics. For example, feature 11066 activates positively for age ($d_z = +0.6$) but negatively for income ($d_z = -1.4$), gender ($d_z = -1.1$), education ($d_z = -1.2$), and vote ($d_z = -1.4$). This pattern indicates encoding of composite social axes where correlated demographic attributes (e.g., younger/wealthier/female/educated/voter) load together, rather than encoding independent demographic dimensions.

**Mean encoding strength is weak relative to early layers.** Across all 50 feature×demographic cells (10 features × 5 demographics), mean $|d| = 0.62$ in Gemma and $|d| = 0.54$ in Qwen, substantially below the early-layer encoding peaks reported in §5.2 (Gemma L9: $|d| = 2.02$; Qwen L15: $|d| = 0.52$). Individual features can show strong encoding for specific demographics (e.g., Qwen feature 44474: gender $d = -2.6$; Gemma feature 15398: gender $d = +1.6$), but these are exceptions. The typical pattern is moderate, distributed encoding across multiple demographics, reinforcing the central finding that the features most causally important for normative outputs are not the features most detectable by standard encoding measures.

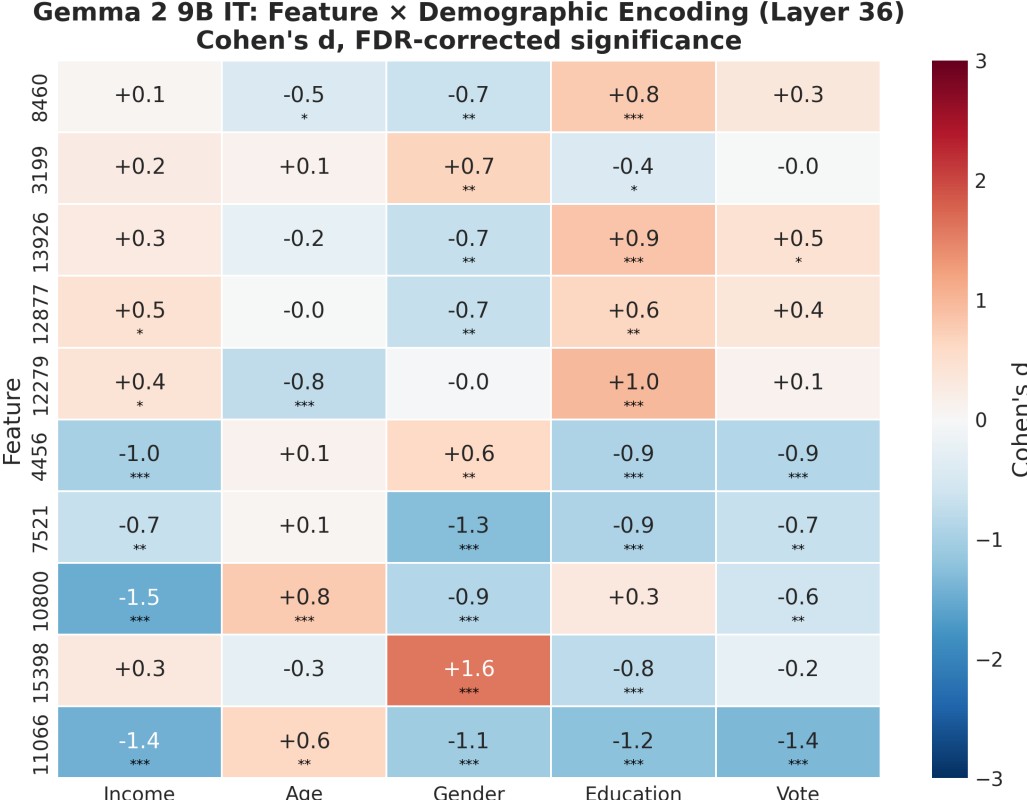

Figure 6: G
emma 2 9B IT: encoding matrix for the 10 most causally important features at L36. Cells show signed Cohen's $d_z$; significance after BH-FDR correction: $^*p < 0.05$, $^{**}p < 0.01$, $^{***}p < 0.001$. Each row is a single SAE feature; columns are the five tested demographic dimensions. The opposite-direction encoding visible across rows indicates that single features encode composite social axes spanning correlated demographic attributes, rather than acting as single-demographic detectors.

This composite-axis structure provides the feature-level mechanism underlying the bundling effect quantified causally in §5.5 and Appendix F: when correlated demographic attributes load on the same features, operations that target one demographic direction necessarily affect the others encoded on the same axis. Per-architecture differences, the Qwen encoding matrix, and the response-position concentration analysis are reported in Appendix D.5.

## 5.7 Cross-Architecture Replication: Llama

We replicate the core analysis on Llama 3.1 8B Instruct across three layers (L15, L23, L27) to test generalizability beyond the primary models. All 25 demographic×domain cells show significant behavioral effects, and 479 pairs remain after floor-effect exclusion. Llama partially replicates the encoding–causal dissociation: mean $|d_{enc}|$ of selected features declines from 4.73 (L15) to 1.19 (L27), yet causal specificity emerges only at L27, where encoding is weakest. Patching at L27 recovers 68.7% of baseline effects against 22.8% for Random-A controls ($d_z = 0.52$, $p < 10^{-26}$), and steering amplifies by 62.3% against 25.8% ($d_z = 0.36$, $p < 10^{-13}$).

Ablation, however, produces only a small effect that does not reach statistical significance ($d_z = 0.08$, $p = 0.07$), in contrast to the strong ablation effects observed in Gemma and Qwen. Llama therefore replicates the encoding side of the dissociation and partially replicates the causal side: patching and steering show clear specificity, but ablation does not eliminate the behavioral effect to a statistically detectable degree.

One possible explanation is distributed rather than concentrated demographic encoding: if information is spread across many features in Llama, then a 50-feature ablation captures only a small fraction of the relevant signal, while patching and steering still succeed because they impose a directional perturbation that propagates through the remaining demographic-relevant features. This hypothesis predicts that sequential ablation of disjoint top-$K$ batches would produce accumulating reductions in the behavioral effect; we have not tested this here and flag it as a direction for future work. Demographic breakdowns and the full encoding profile are reported in Appendix D.4.

## 6 Discussion

The encoding–causal dissociation demonstrates that demographic information undergoes a functional transformation across model depth, and that representational prominence is not a reliable indicator of causal influence on normative outputs.

### 6.1 Representation Versus Deployment

The encoding-causal independence we document is, conceptually, a decomposition of what prior work has called "representational strength" into two empirically distinguishable quantities: *encoding presence* (linear detectability in activations) and *causal mediation* (functional contribution to model output). The implicit assumption in much of the bias-auditing and probing literature is that these track one another: that strong encoding presence licenses behavioral claims, and that erasing encoding presence removes behavioral effects. Our results show that this assumption fails in two distinct ways. Encoding presence is high at layers where causal mediation is near zero, and causal mediation is highest at layers where encoding presence is weak.

In both architectures, early and middle layers construct representations of demographic identity that are causally inert: intervention effects are indistinguishable from random controls, and variance-matched null baselines confirm the early-layer null is genuine rather than an artifact of feature selection. Causal influence emerges only at late layers, with a different representational signature: greater domain specialization and sensitivity to the specific demographic contrast rather than to perturbation in general.

The two models achieve this separation through different encoding strategies. Gemma shows opposing depth gradients, with encoding peaking early and declining as causal influence rises. Qwen shows uniformly weak encoding across all selected features, with causal influence concentrated at the final layer. The invariant is not the encoding profile but the late-layer concentration of causal effects, suggesting that transformers separate demographic representation from its deployment in normative judgment regardless of how the encoding itself is organized. Direct cross-layer transplantation (§5) provides independent evidence that early and late demographic encodings are functionally distinct rather than transformations of one another, supporting encoding–causal independence rather than a unified-but-redistributed representation.

Characterizing the late-layer causal features reveals why this dissociation poses a practical detection problem. Rather than encoding individual demographic attributes cleanly, these features are polysemantic: top causal features in Gemma consistently encode multiple demographics simultaneously, typically with opposite-direction encoding across demographics (e.g., younger/wealthier/educated loading together as a composite social axis). Single-demographic probes applied at these layers would fail to isolate features that encode demographics as correlated bundles rather than independent dimensions (§5.6; Appendix D.5). This polysemantic structure has measurable causal consequences: bootstrapped orthogonalization against correlated demographic encodings (§5.5; Appendix F) confirms that demographic features at late layers carry causal contributions not only from the on-target demographic but also from its correlated structure, with a larger effect in Qwen than in Gemma.

### 6.2 Implications for Bias Auditing

Our findings on polysemantic encoding and encoding–causal dissociation have implications for methods that operate on the assumption of single-direction concept representations. Concept erasure is one such literature: Ravfogel et al. (2020) and Belrose et al. (2025) provide methods that make a target attribute

linearly undetectable in a representation. Two specific implications follow from our findings for methods of this kind, neither of which we test directly.

First, the encoding–causal dissociation we document means that projection-based methods applied at layers with strong demographic encoding (early to middle layers in our models) may remove linear detectability without removing causal influence. The cross-layer transplantation results (§5) show empirically that early-layer demographic encodings carry little causal weight, suggesting that audits or interventions targeting these layers would report success on detectability metrics while leaving behavioral effects intact.

Second, the polysemantic bundling characterized in §5.6 and validated causally in §5.5 means that at late layers, where causal mediation concentrates, demographic information is encoded on composite axes correlated across multiple attributes. Methods that operate on cross-covariance structure between representation and concept span multiple demographic attributes simultaneously when concepts are entangled in this way, meaning single-attribute operations at causally-active layers involve trade-offs across correlated demographic structure rather than clean isolation of a target attribute. The architecture-dependent magnitude of this bundling (approximately 3pp pooled in Gemma, 14pp in Qwen; Appendix F) suggests these trade-offs are not uniform across models. Polysemantic encoding has analogous implications for single-concept probing, steering, and ablation methods that assume clean concept directions.

The domain-contingency of demographic influence compounds this problem. NPE varies substantially across demographic×domain combinations (Gemma IQR (interquartile range): 59% to 89%; Qwen IQR (interquartile range): 79% to 157%), meaning that aggregate fairness evaluations would miss the heterogeneous structure of demographic influence.

For value-sensitive deployments, audits must therefore be both causally validated and domain-specific. These conclusions extend recent findings that models passing explicit bias benchmarks still harbor implicit biases (Bai et al., 2025) and that demographic representations are linear and causal (Bouchaud & Ramaciotti, 2025), by adding that the strength of encoding does not predict causal influence, and that the distribution of influence is uneven across policy areas.

### 6.3  Limitations

Three models in the 7–9B range do not establish that the dissociation holds at larger scales, though consistency across Gemma, Qwen, and Llama with different alignment procedures and SAE regimes suggests it is not idiosyncratic to a single model family. Quantitative details are architecture-dependent: Qwen shows higher random baselines and elevated cross-condition transfer, while Llama shows non-significant ablation ($d_z = 0.08$, $p = 0.07$), consistent with redundant encoding (§5.7). The qualitative pattern, late-layer causal concentration with early-layer encoding, holds across all three.

Our contrastive design uses extreme demographic values to maximise feature identification sensitivity (§3), which is appropriate for the discovery question we address but does not establish that the same features mediate effects under naturalistic demographic variation, non-Western contexts, or open-ended generation. The numeric scoring setup isolates demographic conditioning at the response position, which provides a clean causal target but excludes the inference dynamics that arise with chain-of-thought prefixes, intermediate context propagation, or other open-ended decoding regimes; whether late-layer demographic features remain causally active in those settings is an open question for future work.

Magnitude estimates for prompts where the model's answer distribution concentrates at the scale boundaries (values 1 and 10 on 11-point scales) are sensitive to the choice of scoring method for the ambiguous first token "1". Sequence-level scoring confirms the encoding–causal dissociation is robust in aggregate (pooled recovery shift $-1.9$pp, 95% CI $[-6.1, +1.3]$), but vote × values-domain patching recovery specifically shifts by approximately $-32$pp under sequence-level scoring, reflecting the boundary-concentrated answer geometry of this cell. Magnitude estimates for this specific subset should be interpreted with corresponding uncertainty (see Appendix D.2).

The magnitude estimate of polysemantic bundling carries methodological uncertainty. Bootstrapped orthogonalization against correlated demographic encodings (Appendix F) yields wide 95% confidence intervals for Qwen specifically (pooled bundling residual 14.1pp $[7.0, 30.9]$), reflecting that encoding-direction esti-

mates from $N = 30$ held-out pairs carry meaningful sampling variance in this architecture. The directional conclusion—positive bundling in every bootstrap resample is robust; the precise magnitude is uncertain by approximately $\pm 15$pp.

All models studied are instruction-tuned, leaving open whether the dissociation reflects architectural properties or is induced by alignment training. Base-versus-instruction-tuned comparisons would address this directly.

Finally, most causally important features lack clear semantic labels, limiting interpretation beyond the causal-functional level.

## 7 Conclusion

We presented a mechanistic analysis of how demographic context causally influences normative reasoning in large language models. Using sparse autoencoders and three causal interventions across Gemma 2 9B, Qwen 2.5 7B, and Llama 3.1 8B, we traced the path from demographic encoding to demographic influence and found that these are distinct phenomena occurring at different network depths. Early layers construct representations of demographic identity that contribute minimally to outputs, while late layers, where encoding is weaker, carry the causal leverage that shapes normative judgments. The dissociation holds across two architectures with partial replication on Llama (encoding side replicated, ablation not statistically detectable), under different encoding profiles and alignment procedures, and is further modulated by policy domain.

The practical implication is that bias auditing strategies relying on representational detection will surface the wrong model components, targeting features that are visible but causally inert while missing the late-layer, domain-specific features that actually drive outputs. Effective governance of language models in value-sensitive applications requires causal validation of representational findings, conducted separately for each deployment domain.

**Broader Impact Statement**

Our methods are designed for bias auditing, but the same pipeline could in principle be used to amplify rather than mitigate demographic influence; for example, by identifying late-layer causal features and steering them to increase demographic sensitivity in targeted domains. We believe the transparency value of publishing the methodology outweighs this risk, as effective bias mitigation requires the research community to understand the mechanisms it seeks to govern, and because the interventions we describe require white-box access to model weights, limiting misuse by external actors.

The European Social Survey data used in this study are collected under established ethical protocols with informed consent from participants, governed by ESS ERIC's data protection framework in compliance with GDPR, and made available for scientific research under a registered user agreement. We use only aggregate response patterns and do not attempt to identify individual respondents. The binary gender contrast in our experimental design is constrained by the ESS panel variable and does not reflect the full spectrum of gender identities; we flag this as a limitation and note that extending the framework to non-binary identities is an important direction for future work.

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

## A    Transfer Error Definitions

The main text defines three normalized intervention metrics: NPE for patching (Equation 6), NAE for ablation (Equation 7), and NSE for steering (Equation 8), each expressing effect size relative to $\Delta_{\text{EV}}$. Normalization enables comparison across scales but can obscure interpretation when baseline effects are small or when interventions overshoot their targets. We therefore report signed and absolute transfer errors for all three interventions, expressed in original scale units.

Each intervention has a target outcome. The signed transfer error (STE) is the difference between the observed post-intervention behavioral gap and this target; |STE| is its magnitude.

**Patching.**    The target is exact reproduction of condition $a$'s baseline output, i.e., $\text{EV}_b' = \text{EV}_a$, yielding a post-intervention gap of $\Delta_{\text{EV}}$:

$$\text{STE}_{\text{patch}} = (\text{EV}_b' - \text{EV}_b) - \Delta_{\text{EV}} \tag{9}$$

**Ablation.**    The target is complete elimination of the demographic effect, i.e., $\text{EV}_a' = \text{EV}_b$, yielding a post-intervention gap of zero:

$$\text{STE}_{\text{ablation}} = \text{EV}_a' - \text{EV}_b \tag{10}$$

**Steering.**    The target is NSE by factor $\alpha$ ($= 2.0$), i.e., a post-intervention gap of $\alpha\Delta_{\text{EV}}$:

$$\text{STE}_{\text{steer}} = (\text{EV}_a' - \text{EV}_b) - \alpha\Delta_{\text{EV}} \tag{11}$$

For all three, STE $= 0$ indicates the intervention achieved its target exactly. Negative values indicate undershoot and positive values indicate overshoot. |STE| treats both directions symmetrically. Table 1 summarises the metrics.

Table 1: normalized metrics and transfer errors for each intervention. Perfect transfer corresponds to STE $= 0$ and the normalized metric equalling 1.0.

| Intervention | normalized metric | Target gap | STE definition | Perfect transfer |
|---|---|---|---|---|
| Patching | NPE (Eq. 6) | $\Delta_{\text{EV}}$ | $(\text{EV}_b' - \text{EV}_b) - \Delta_{\text{EV}}$ | STE $= 0$, NPE $= 1$ |
| Ablation | NAE (Eq. 7) | $0$ | $\text{EV}_a' - \text{EV}_b$ | STE $= 0$, NAE $= 1$ |
| Steering | NSE (Eq. 8) | $\alpha\Delta_{\text{EV}}$ | $(\text{EV}_a' - \text{EV}_b) - \alpha\Delta_{\text{EV}}$ | STE $= 0$, NSE $= 1$ |

## B    Causal Validation Details

This appendix provides detailed results supporting the causal intervention analysis in §5. All means use 1st/99th percentile Winsorization; steering means use a $\pm 200\%$ cap (§4.4).

### B.1    Main Intervention Results

Tables 2 and 3 report metrics at the best layer ($K = 50$, per-pair). All interventions significantly exceed Random-A controls ($p < 0.001$).

Table 2: Gemma 2 9B IT at L36 ($K = 50$, per-pair).

|  | Mean | Med. | Rnd-A | $d_z$ | $N$ |
|---|---|---|---|---|---|
| NPE | 75.8 | 90.7 | 26.7 | 0.99 | 384 |
| NSE | 65.1 | 40.6 | 42.8 | 0.44 | 384 |
| NAE | 74.5 | 85.1 | 26.6 | 1.27 | 384 |

Table 3: Qwen 2.5 7B IT at L27 ($K=50$, per-pair).

|  | Mean | Med. | Rnd-A | $d_z$ | $N$ |
|---|---|---|---|---|---|
| NPE | 132.9 | 102.3 | 53.2 | 0.63 | 364 |
| NSE | 75.3 | 69.6 | 13.2 | 0.67 | 364 |
| NAE | 62.5 | 96.3 | $-23.3$ | 0.56 | 364 |

## B.2 Aggregate vs. Per-Pair Selection

Per-pair selection chooses the $K$ most active features per contrastive pair (upper bound). Aggregate selection uses population-level top-$K$ features (§4.3), testing generalisation. In Gemma ($N_{\mathrm{agg}}=97$), aggregate features retain most of the per-pair effect (ratio $\geq 0.70$). In Qwen ($N_{\mathrm{agg}}=260$), aggregate features perform substantially worse (patching ratio 0.27), indicating more pair-specific features.

Table 4: Gemma: agg. vs. per-pair, L36. $N_{\mathrm{agg}}=97$.

|  | Per-Pair | Agg. | Ratio |
|---|---|---|---|
| NPE | 75.8 | 55.4 | 0.73 |
| NSE | 65.1 | 70.1 | 1.08 |
| NAE | 74.5 | 52.2 | 0.70 |

Table 5: Qwen: agg. vs. per-pair, L27. $N_{\mathrm{agg}}=260$.

|  | Per-Pair | Agg. | Ratio |
|---|---|---|---|
| NPE | 132.9 | 36.3 | 0.27 |
| NSE | 75.3 | 13.6 | 0.18 |
| NAE | 62.5 | $-5.3$ | $-0.08$ |

## B.3 Demographic Breakdown

Tables 6 and 7 disaggregate results by demographic (mean / median). In Gemma, education shows the highest NPE (90.3) and gender the lowest (62.7). In Qwen, gender shows the largest mean–median divergence for patching (209.3 vs. 109.7), reflecting overshoot on a subset of pairs.

## B.4 Dose-Response

Figures 7 and 8 show intervention effects as a function of $K$ at the best layer. In Gemma at L36, all three interventions increase monotonically: NPE from 39.8 ($K=5$) to 75.8 ($K=50$); NSE from 47.9 to 65.1; NAE from 40.8 to 74.5. In Qwen at L27, NPE rises from 75.8 ($K=5$) to 132.9 ($K=50$), saturating by $K=100$ (132.7). NSE rises from 40.5 to 75.3; NAE from 5.5 to 62.5 (both saturating at $K=100$). We tested $K=100$ only for Qwen. In both models, the specificity gap widens with $K$.

## B.5 Floor-Effect Exclusion

We exclude pairs with $|\Delta_{\mathrm{EV}}|$ below a scale-normalized threshold (0.3 scale points for range $=10$, scaled proportionally: 0.03 for range $=1$, 0.09 for range $=3$, 0.12 for range $=4$). After exclusion, 384 pairs remain for Gemma (L36) and 364 for Qwen (L27).

Table 6: Gemma: by demographic, L36. Mean / med.

|      | $N$ | NPE | NSE | NAE |
|------|-----|-----|-----|-----|
| Inc. | 102 | 76.7/94.3 | 49.1/24.8 | 80.9/96.1 |
| Age  | 42  | 81.3/88.1 | 68.6/42.3 | 89.1/92.2 |
| Gen. | 67  | 62.7/76.8 | 79.4/49.9 | 58.2/69.2 |
| Edu. | 81  | 90.3/98.8 | 74.5/75.2 | 71.4/81.9 |
| Vote | 92  | 69.1/90.2 | 62.3/23.4 | 75.3/77.1 |

Table 7: Qwen: by demographic, L27. Mean / med.

|      | $N$ | NPE | NSE | NAE |
|------|-----|-----|-----|-----|
| Inc. | 105 | 114.0/101.4 | 77.1/52.9 | 61.4/100.0 |
| Age  | 62  | 108.5/93.0 | 79.3/80.6 | 55.4/93.8 |
| Gen. | 42  | 209.3/109.7 | 94.6/113.0 | 15.5/82.6 |
| Edu. | 75  | 167.0/111.6 | 81.3/77.6 | 44.7/92.7 |
| Vote | 80  | 104.8/95.2 | 54.2/34.4 | 110.5/96.2 |

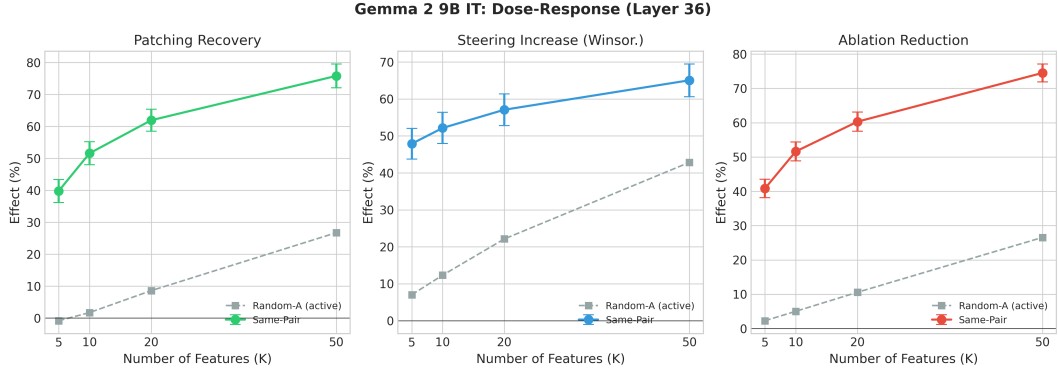

Figure 7: Gemma 2 9B IT: dose-response at Layer 36. All three interventions increase monotonically with $K$, with specificity gaps widening at each step.

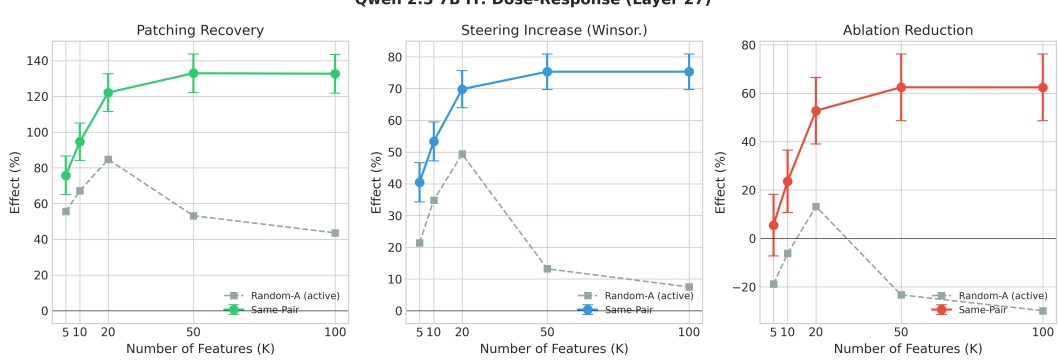

Figure 8: Qwen 2.5 7B IT: dose-response at Layer 27. All three interventions saturate by $K = 50$; $K = 100$ confirms no additional gains.

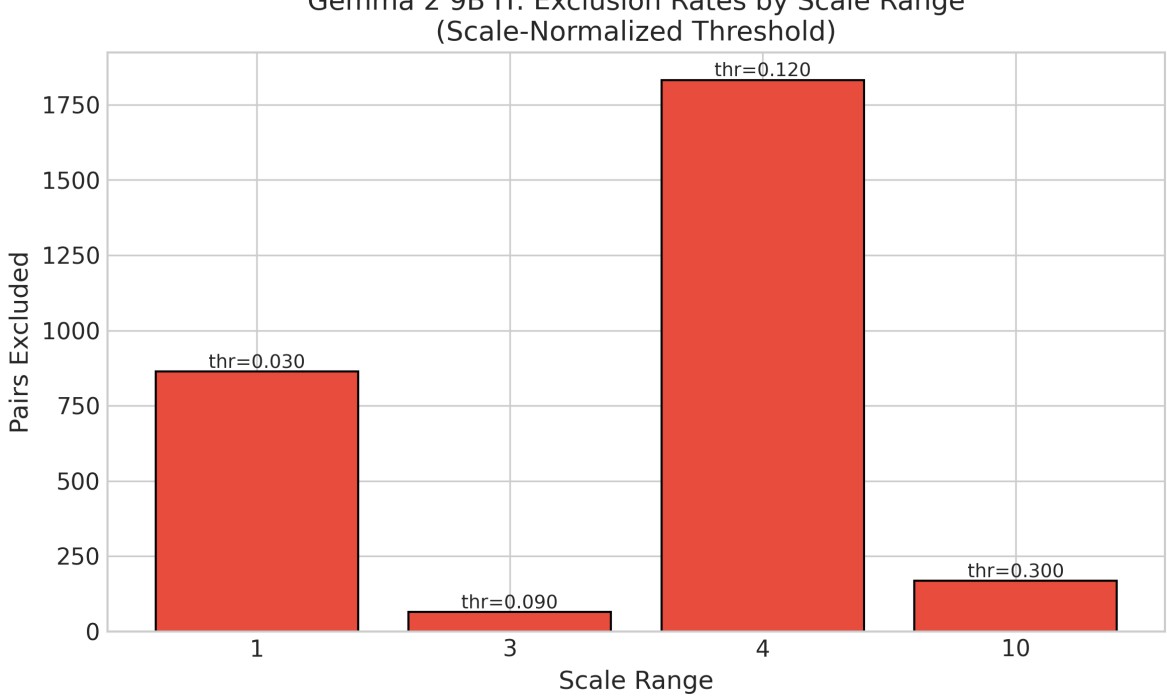

Figure 9: Gemma: pairs excluded by scale range.

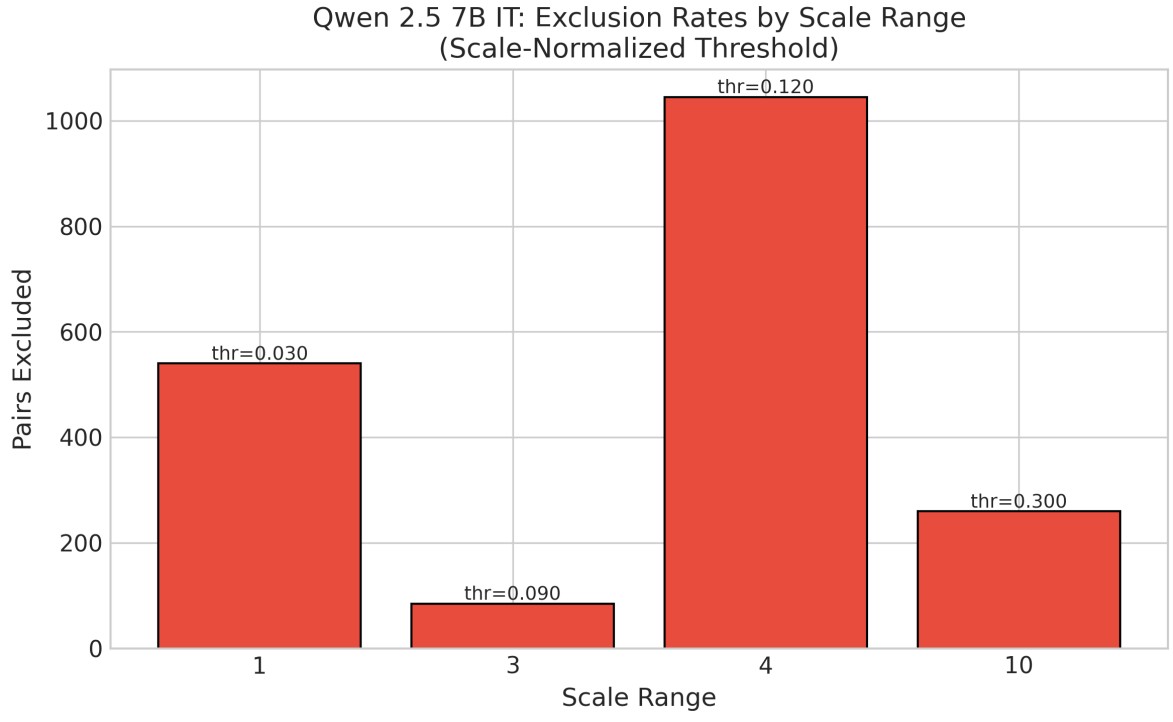

Figure 10: Qwen: pairs excluded by scale range.

# C   Feature Identification Details

## C.1   Filter Pipeline

From the full set of $m = 16{,}384$ SAE features at each layer, we select demographic-relevant features through a three-stage pipeline applied independently to each demographic×domain group (25 groups per layer).

**Stage 1: Noise filtering.**   We exclude features with inconsistent activation differences across contrastive pairs. A feature passes if it shows consistent sign direction (sign agreement $\geq 0.6$)) and is not flagged as noisy based on coefficient of variation and outlier ratio thresholds.

**Stage 2: Statistical testing.**   We apply one-sample $t$-tests on mean activation differences within each group, with Benjamini-Hochberg correction at $\alpha = 0.05$.

**Stage 3: Effect size thresholding.**   We retain features with $|\text{Cohen's } d_z| \geq 0.3$. From the features passing all three stages, we select the top $K = 50$ by mean absolute activation difference.

## C.2   Feature Yield

Table 8 reports the number of features passing all filter stages at each layer, averaged across the 25 demographic×domain groups. In Gemma, feature yield peaks at middle layers (L14, L18) and declines at late layers where causal effects concentrate. In Qwen, yield increases with depth, with the most features at L27 (the causally important layer). This pattern is consistent with the dissociation: layers with many detectable features are not necessarily the layers where those features causally matter. Some Qwen demographic×domain groups yield as few as 1 feature after filtering, reflecting the uniformly weak encoding in that model. Causal validation uses per pair feature selection, which is not constrained by population-level yield. The mean $|d_{\text{enc}}|$ of selected features declines with depth in Gemma (2.02 at L9 to 0.99 at L36),

Table 8: Feature yield per layer. Mean (min–max) selected features per demographic×domain group, and mean $|d_{\text{enc}}|$ of all selected features at each layer.

| Layer | Mean/group | Range | $|d_{\text{enc}}|$ |
|---|---|---|---|
| *Gemma 2 9B IT* | | | |
| L5 (12%) | 37.9 | 30–48 | 1.78 |
| L9 (22%) | 30.8 | 24–37 | 2.02 |
| L14 (34%) | 47.5 | 36–50 | 2.00 |
| L18 (44%) | 48.0 | 38–50 | 1.96 |
| L20 (49%) | 25.8 | 15–34 | 1.42 |
| L27 (66%) | 46.3 | 31–50 | 1.15 |
| L32 (78%) | 34.4 | 17–50 | 1.02 |
| L36 (88%) | 32.8 | 12–50 | 0.99 |
| *Qwen 2.5 7B IT* | | | |
| L7 (26%) | 52.8 | 1–100 | 0.41 |
| L15 (56%) | 29.4 | 1–100 | 0.52 |
| L19 (70%) | 41.7 | 1–100 | 0.43 |
| L23 (85%) | 60.6 | 1–100 | 0.32 |
| L27 (100%) | 72.4 | 1–100 | 0.31 |

consistent with the encoding gradient reported in §5.2. In Qwen, encoding is weaker overall ($|d_{\text{enc}}| \leq 0.52$) and relatively flat across layers. Figures 11 and 12 show the full encoding strength profile disaggregated by demographic×domain.

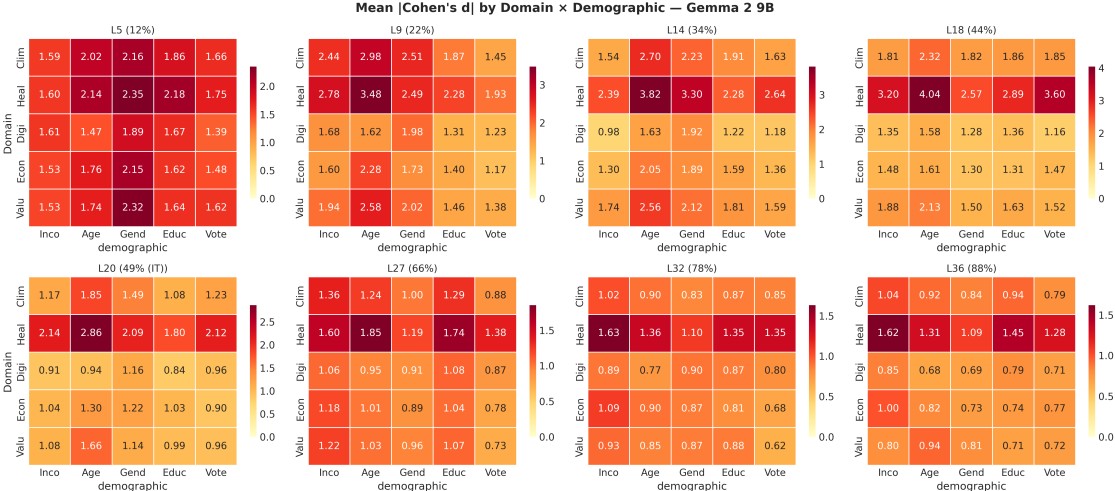

Figure 11: Gemma 2 9B IT: mean $|d_{\mathrm{enc}}|$ of selected features by domain×demographic at each layer. Encoding peaks at early-to-middle layers and declines with depth across all cells.

### C.3 Encoding by Domain and Demographic

Figures 11 and 12 show mean |Cohen's $d$| of selected features for each demographic×domain cell at every layer. Several patterns are notable.

In Gemma, encoding is strongest for age×health (peaking at $|d_{\mathrm{enc}}| = 4.04$ at L18) and gender across most domains. Encoding declines monotonically with depth for all cells, but the rate varies: health-related cells retain stronger encoding at late layers than digital governance or economy cells. At L36, health cells still show $|d_{\mathrm{enc}}| > 1.0$ while most other cells fall below 1.0.

In Qwen, encoding is substantially weaker across the board ($|d_{\mathrm{enc}}|$ rarely exceeds 1.0). Vote×climate shows the strongest early encoding ($|d_{\mathrm{enc}}| = 1.27$ at L7). Gender shows relatively high encoding at middle layers across domains (e.g., gender×values: 0.60 at L27). The flatter encoding profile in Qwen contrasts with Gemma's steep decline but, as shown in §5.2, both models exhibit the same late-layer concentration of causal effects.

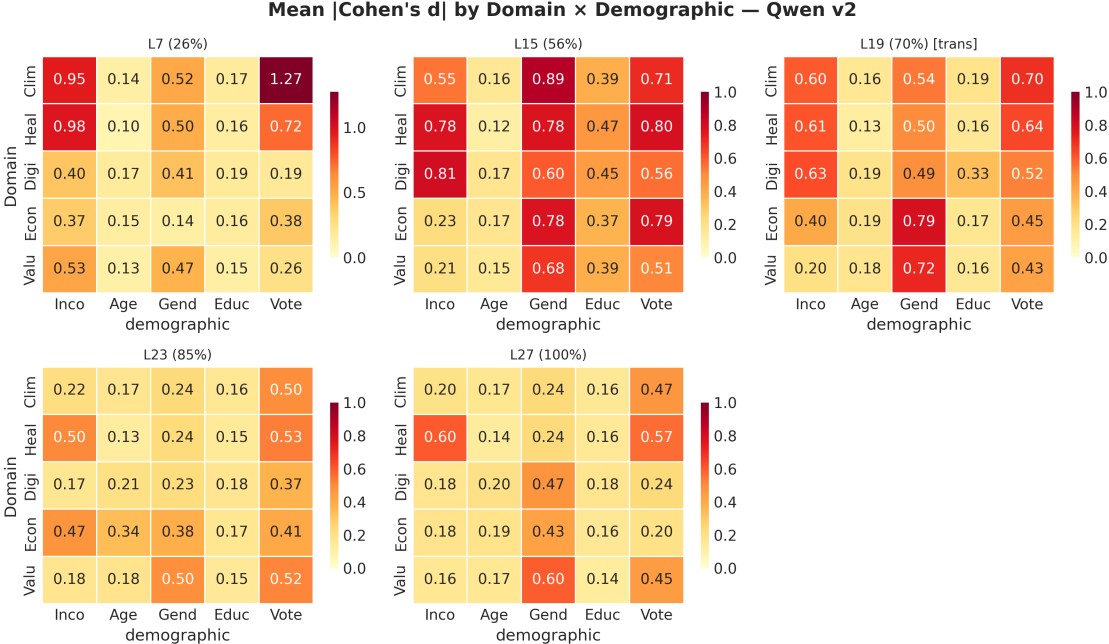

Figure 12: Qwen 2.5 7B IT: mean $|d_{\mathrm{enc}}|$ by domain×demographic at each layer. Encoding is weaker overall and relatively flat across depth.

## D  Technical Validation and behavioral Baseline

This appendix reports SAE reconstruction quality (§D.1), tokenizer handling (§D.2), and full behavioral baseline results (§D.3).

### D.1  SAE Reconstruction Quality

We use width-16K Gemma Scope SAEs (Lieberum et al., 2024) for Gemma 2 9B and width-131K SAEs from Gao et al. (2024) for Qwen 2.5 7B. Table 9 reports reconstruction quality for Gemma across all eight analyzed layers; Table 10 reports the same for Qwen across five layers.

For Gemma, cosine similarity ranges from 0.82 (L32) to 0.91 (L5), with NMSE between 0.17 and 0.32. Reconstruction quality degrades modestly at late layers (L27–L36), coinciding with higher observed L0 sparsity which refers to the number of active features per input, (97–112 active features vs. the canonical 61–77). All layers pass standard quality thresholds (cosine > 0.80).

For Qwen, cosine similarity ranges from 0.82 (L27) to 0.96 (L3) on demographic prompts. Reconstruction is weaker at L23 and L27 (cosine < 0.85). However, for our differential analysis (contrastive activation differences), reconstruction errors cancel to first order if they are consistent across conditions. Per-prompt cosine similarity exceeds 0.90 for most individual prompts even at L15 (Table 11), suggesting that the SAE captures condition-relevant variation well despite moderate aggregate error.

L20 for Gemma uses the instruction-tuned (IT) SAE variant; all other Gemma layers use pretrained (PT) SAEs. All Qwen SAEs are IT variants.

**Per-prompt reconstruction (Qwen L15).**  Table 11 shows that individual demographic prompts achieve cosine $\geq 0.86$ at L15, confirming adequate reconstruction on the prompts used in our pipeline. Tables 9 and 10 report different reconstruction metrics because the two SAE frameworks (Gemma Scope and SAELens) expose different default diagnostics. Cosine similarity and sparsity (L0) are directly comparable across both.

Table 9: Gemma SAE reconstruction quality per layer. $R^2$ denotes explained variance $(1 - \mathrm{NMSE})$, distinct from the expected value EV defined in Equation 3.

| Layer | SAE | NMSE | Cos | $R^2$ | L0 | Alive |
|-------|-----|------|------|------|-----|-------|
| L5 | PT | 0.17 | 0.91 | 0.83 | 64 | 0.5% |
| L9 | PT | 0.23 | 0.88 | 0.77 | 44 | 0.4% |
| L14 | PT | 0.24 | 0.87 | 0.76 | 70 | 0.8% |
| L18 | PT | 0.22 | 0.89 | 0.78 | 65 | 0.9% |
| L20 | IT | 0.20 | 0.89 | 0.80 | 38 | 0.6% |
| L27 | PT | 0.27 | 0.85 | 0.73 | 112 | 3.8% |
| L32 | PT | 0.32 | 0.82 | 0.68 | 97 | 3.4% |
| L36 | PT | 0.28 | 0.85 | 0.72 | 104 | 4.2% |

PT = pretrained SAE; IT = instruction-tuned SAE.

Table 10: Qwen SAE reconstruction quality per layer (demographic prompts only).

| Layer | Cos | RelErr | L0 | L1 |
|-------|------|--------|-----|------|
| L3 (11%) | 0.96 | 27.0% | 50 | 79.8 |
| L7 (26%) | 0.91 | 41.0% | 64 | 154.0 |
| L11 (41%) | 0.91 | 41.7% | 58 | 190.3 |
| L15 (56%) | 0.90 | 42.9% | 56 | 225.4 |
| L19 (70%) | 0.88 | 47.4% | 51 | 298.4 |
| L23 (85%) | 0.85 | 52.7% | 54 | 629.8 |
| L27 (100%) | 0.82 | 57.5% | 52 | 1601.3 |

### D.2 Tokenizer Handling

Our experimental design requires models to select responses from a fixed set of numerical options presented on Likert-type and categorical scales. Scale endpoints vary across ESS items (e.g., 0–10 for agreement, 1–4 for categorical). Two tokenization issues arise.

**Multi-token numbers.** The value "10" tokenizes to two tokens in both Gemma and Qwen tokenizers ("1" + "0"). Since the model predicts one token at a time, the first-token probability $P(\text{"1"})$ conflates two response values: 1 and the first digit of 10. For 11-point scales (0–10), we split $P(\text{"1"})$ equally between value 1 and value 10 (i.e., $P(1) = P(10) = \frac{1}{2}P(\text{"1"})$).

**Expected value computation.** For each item, we extract logits over the full vocabulary at the response position, restrict to valid scale tokens (including multi-token handling as described above), renormalize via softmax to obtain a probability distribution over valid responses, and compute the expected value EV $= \sum_v v \cdot P(v)$.

**Sequence-level scoring sensitivity.** The equal-split heuristic is a conservative approximation; a more principled alternative computes $P(v{=}1)$ and $P(v{=}10)$ as joint sequence probabilities via a second forward pass conditioned on the first token:

$$P(v{=}1) = P(\text{"1"} \mid \text{prompt}) \cdot P(\text{EOS-like} \mid \text{prompt} + \text{"1"}) \tag{12}$$

$$P(v{=}10) = P(\text{"1"} \mid \text{prompt}) \cdot P(\text{"0"} \mid \text{prompt} + \text{"1"}) \tag{13}$$

where the EOS-like set comprises sentence terminators, whitespace, and the chat template `<end_of_turn>` token. For patched conditions the second-token distribution is computed under the same residual-stream patch as the first token.

We validated the equal-split heuristic against sequence-level scoring on 240 stratified pairs (12 per demographic × domain cell, scale-11 questions only) from Gemma 2 9B at L36. The two methods agree closely on

Table 11: Qwen per-prompt reconstruction at L15.

| Prompt | Cos | RelErr | L0 |
|---|---|---|---|
| wealthy/F/FR/climate | 0.92 | 40.0% | 54 |
| poor+old/M/DE/health | 0.90 | 43.4% | 47 |
| young+poor/F/PL/econ | 0.89 | 44.7% | 53 |
| wealthy(A)/M/UK/health | 0.91 | 40.4% | 49 |
| poor(B)/M/UK/health | 0.91 | 40.4% | 51 |
| old+nonvote/F/SE/values | 0.94 | 34.0% | 23 |
| young+edu/M/NL/digital | 0.87 | 49.3% | 76 |
| mid+lowed/F/ES/econ | 0.86 | 51.1% | 91 |

per-prompt EV (Pearson $r = 0.97$, Spearman $\rho = 0.97$ on $N = 480$ prompts; mean absolute difference 0.085 on the 0–10 scale), on contrastive $\Delta$EV ($r = 0.97$, sign agreement 95.8%), and on pooled patching recovery (original 67.7%, sequence-level 65.8%, mean shift $-1.9$pp [95% bootstrap CI: $-6.1, +1.3$] on $N = 192$ pairs above the effect threshold). The pooled CI crosses zero, and four of five demographics individually show CIs crossing zero (income $+0.7$pp [$-0.2, +2.0$], age $+4.3$pp [$-0.0, +13.0$], gender $+0.1$pp [$-0.0, +0.2$], education $+1.1$pp [$-2.2, +5.8$]). Vote is the exception ($-11.6$pp [$-28.2, -0.4$]), with the shift essentially confined to the values domain.

A focused vote-only re-analysis with proper patched second-token scoring ($N = 46$ vote pairs) gave a pooled vote shift of $-7.7$pp [$-18.4, -0.3$]. Per-domain breakdown isolated the shift to vote $\times$ values questions: climate $-3.4$pp, digital $+0.3$pp, economy $+0.0$pp, values $-31.9$pp ($N = 10$). The values-domain questions (e.g., importance of democracy, individual versus state responsibility, civic participation attitudes) produce answer distributions that pile probability heavily at scale boundaries, where the 1/10 token disambiguation has greatest leverage on EV. For pairs where the model's answer distribution is concentrated at boundaries, the equal-split heuristic mis-allocates probability mass in a way that sequence-level scoring corrects. The encoding–causal dissociation finding is therefore robust to scoring-method choice in aggregate and across four of five demographics; magnitude estimates for the vote $\times$ values cell specifically carry approximately $\pm 10$–$30$pp scoring-method uncertainty.

Note that the baseline pooled recovery in this sensitivity analysis (67.7%) is computed on the restricted scale-11 subset ($N = 192$); the main-paper L36 patching recovery (75.8%, $N = 384$ in §5) reflects the full per-pair patching analysis across all scale formats.

### D.3 Behavioral Baseline

Tables 12 and 13 report the full behavioral effect for each demographic$\times$domain cell. Effects are computed as the mean EV difference between contrastive persona pairs (condition $a$ minus condition $b$), averaged across all items within each domain. We report significance at $p < 0.05$ (paired $t$-test).

Gemma shows significant effects in 24 of 25 cells, with the non-significant cell being gender$\times$digital ($\Delta$EV $= 0.007$, $p > 0.05$). Qwen shows significant effects in 17 of 25 cells, with substantially smaller effects across all cells (max $|\Delta$EV$| = 0.139$ vs. Gemma's 0.594).

Table 12: Gemma 2 9B IT: behavioral effects ($\Delta$EV) by demographic$\times$domain. Bold indicates $p < 0.05$.

| | Inc. | Age | Gen. | Edu. | Vote |
|---|---|---|---|---|---|
| Clim. | **−.476** | **.163** | **.172** | **−.480** | **−.352** |
| Heal. | **−.361** | **−.043** | **.301** | **−.522** | **−.517** |
| Digi. | **.420** | **.073** | .007 | **.084** | **.156** |
| Econ. | **.594** | **.182** | **−.146** | **.153** | **.052** |
| Valu. | **.287** | **−.272** | **−.079** | **.250** | **.158** |

Table 13: Qwen 2.5 7B IT: behavioral effects ($\Delta$EV) by demographic×domain. Bold indicates $p < 0.05$.

|       | Inc.    | Age     | Gen.    | Edu.    | Vote    |
|-------|---------|---------|---------|---------|---------|
| Clim. | .000    | **−.040** | **−.047** | .008    | **−.049** |
| Heal. | **.040** | **.040** | **.029** | −.021   | −.003   |
| Digi. | **−.023** | −.001   | **−.007** | **−.055** | **.009** |
| Econ. | **.139** | .024    | .001    | **−.057** | **−.072** |
| Valu. | **.052** | −.015   | **−.080** | **.069** | **.031** |

**Scale effects.** The magnitude of behavioral effects correlates with scale range: 11-point scales (0–10) tend to show larger absolute $\Delta$EV than 5-point Likert scales (1–5). This motivates the scale-normalized floor-effect exclusion threshold used in the causal analysis (Appendix B.5).

## D.4 Cross-Architecture Replication: Llama 3.1 8B

We replicate the core analysis on Llama 3.1 8B Instruct (three layers: L15, L23, L27) to test cross-model generalizability. All 25 demographic×domain cells show significant behavioral effects; $N = 479$ pairs remain after floor-effect exclusion.

Llama replicates the encoding-causal dissociation: mean $|d_{enc}|$ of selected features declines from 4.73 (L15) to 1.98 (L23) to 1.19 (L27), yet causal effects concentrate at L27 where encoding is weakest.

Table 14 reports causal validation at L27 ($K = 50$, per-pair). Patching and steering exceed Random-A controls (NPE 68.7% vs. 22.8%; NSE 62.3% vs. 25.8%). Ablation is weaker (NAE 29.9% vs. 24.3%, $d_z = 0.08$), consistent with redundant encoding. Gender shows the strongest patching (88.6%) and steering (102.8%); age shows negligible ablation (1.0%).

Table 14: Llama at L27 ($K$=50, $N$=479). Top: aggregate. Bottom: by demographic (mean / med.).

|     | Mean | Med. | Rnd-A | $d_z$ |     | $N$ | NPE | NSE | NAE |
|-----|------|------|-------|-------|-----|-----|-----|-----|-----|
|     |      |      |       |       | Inc. | 127 | 59.3/69.3 | 43.4/52.1 | 41.2/63.8 |
| NPE | 68.7 | 73.5 | 22.8  | 0.52  | Age | 71 | 76.7/69.8 | 61.4/76.1 | 1.0/8.1 |
| NSE | 62.3 | 65.6 | 25.8  | 0.36  | Gen. | 60 | 88.6/85.2 | 102.8/109.4 | 34.9/45.5 |
| NAE | 29.9 | 50.2 | 24.3  | 0.08  | Edu. | 96 | 56.6/63.2 | 67.3/72.0 | 26.0/43.7 |
|     |      |      |       |       | Vote | 125 | 73.2/85.8 | 58.6/54.7 | 35.5/53.6 |

## D.5 Feature Interpretability

This appendix provides detailed per-architecture breakdowns of the composite social axis analysis summarised in §5.6 in the main paper. The feature-level structure documented here is the proposed mechanism underlying the polysemantic bundling effect quantified causally in Appendix F: if causally important features encode multiple demographics simultaneously, then per-pair top-K patching will necessarily transplant correlated demographic information.

To characterize what late-layer causal features encode, we analyze the 10 features with highest mean causal effect at the best layer in each model (Gemma L36, Qwen L27). For each feature, we test encoding of all five demographic dimensions using paired $t$-tests on 30 held-out contrastive pairs per dimension, with Benjamini-Hochberg correction at $\alpha = 0.05$. We report Cohen's $d_z$ as the effect size. Figures 13 and 14 show the full encoding matrices.

**Polysemantic demographic encoding.** In both models, causally important features encode multiple demographics simultaneously rather than acting as single-demographic detectors. In Gemma, all 10 features encode two or more demographics significantly (mean 3.3 per feature); in Qwen, 7 of 10 do (mean 3.1). This polysemanticity explains why single-demographic probes applied at late layers would fail to identify these features: they do not cleanly separate along any one demographic axis. The orthogonalization analysis in Appendix F provides direct causal validation of this account: removing off-target demographic directions

from the per-pair perturbation vector reduces pooled NPE recovery by 3.2pp in Gemma and 14.1pp in Qwen, with effect sizes that track the polysemanticity reported here.

**Composite social axes.** In Gemma, 9 of 10 multi-demographic features show opposite-direction encoding across demographics. For example, feature 11066 activates positively for age ($d_z = +0.6$) but negatively for income ($d_z = -1.4$), gender ($d_z = -1.1$), education ($d_z = -1.2$), and vote ($d_z = -1.4$). This pattern indicates encoding of composite social axes where correlated demographic attributes (e.g., younger/wealthier/female/educated/voter) load together, rather than encoding independent demographic dimensions. In Qwen, the pattern is present but weaker: 4 of 7 multi-demographic features show opposite-direction encoding, with 3 showing same-direction encoding, consistent with the less domain-specific feature structure reported in §5.5.

**Response-position concentration.** In both models, feature activations concentrate at the final (response) token position rather than being distributed across the prompt span. In the majority of feature×demographic cells, span activation is zero, yielding an undefined ratio; among the cells with non-zero span activation, the mean ratio of last-token to span activation magnitude is 82.3× in Gemma (6 of 50 cells) and 56.4× in Qwen. This is consistent with demographic information being integrated at the point of output generation, not during initial prompt processing, though the small number of cells with finite ratios warrants caution.

**Weak per-feature encoding at causal layers.** Mean encoding strength across all 50 feature×demographic cells (10 features × 5 demographics) is $|d| = 0.62$ in Gemma and $|d| = 0.54$ in Qwen, substantially below the early-layer encoding peaks reported in §5.2 (Gemma L9: $|d| = 2.02$; Qwen L15: $|d| = 0.52$, Table 8). Individual features show strong encoding for specific demographics (e.g., Qwen feature 44474: gender $d = -2.6$; Gemma feature 15398: gender $d = +1.6$), but these are exceptions. The typical pattern is moderate, distributed encoding across multiple demographics, reinforcing the central finding that the features most causally important for normative outputs are not the features most detectable by standard encoding measures.

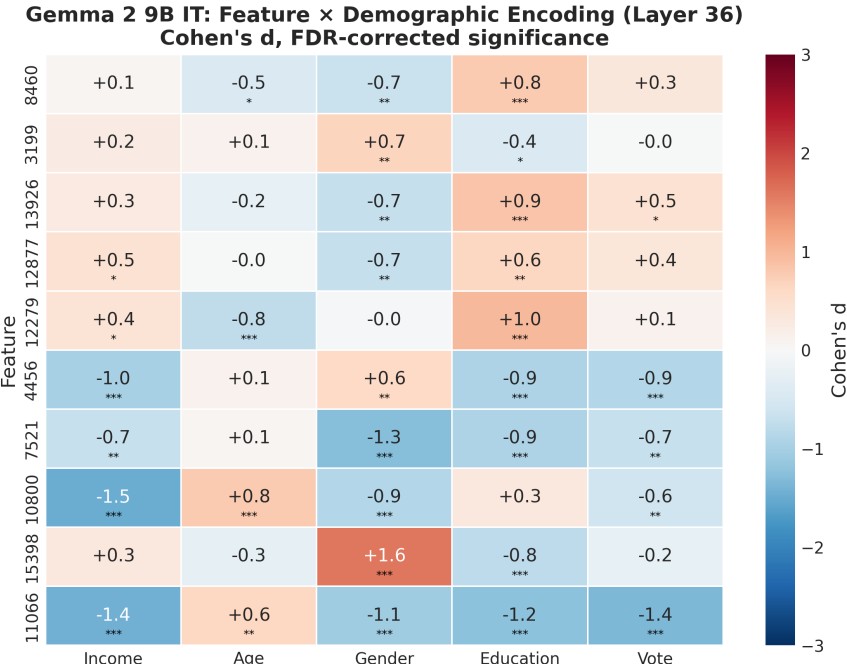

Figure 13: Gemma 2 9B IT: encoding matrix for the 10 most causally important features at L36. Cells show signed Cohen's $d_z$; significance after BH-FDR (False Discovery Rate) correction: $^*p < 0.05$, $^{**}p < 0.01$, $^{***}p < 0.001$.

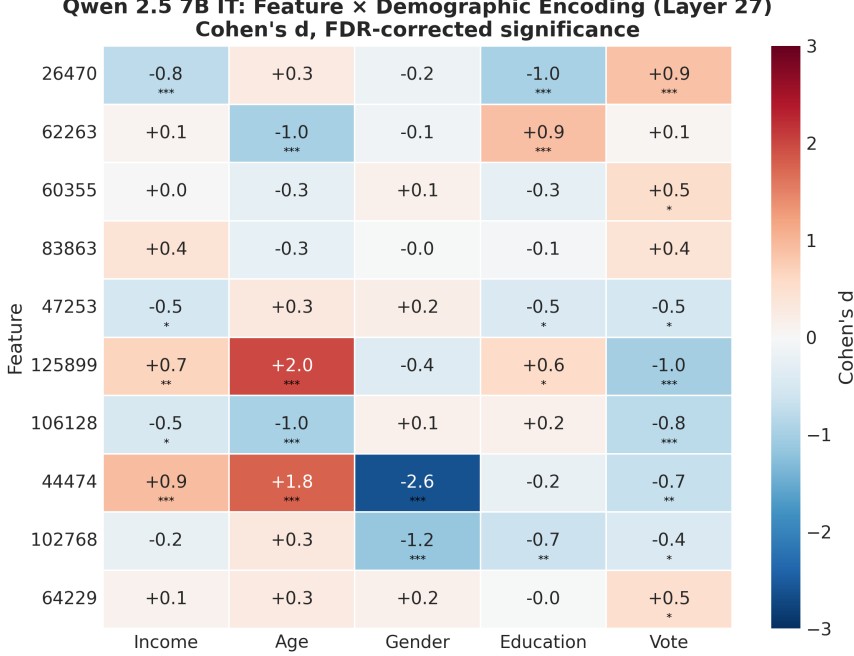

Figure 14: Qwen 2.5 7B IT: encoding matrix for the 10 most causally important features at L27. Conventions as in Figure 13. Encoding is sparser (48% vs. 66% significant cells) and effect sizes are smaller on average, consistent with Qwen's weaker overall demographic sensitivity.

# E  Cross-Layer Transplantation Details

This appendix details the cross-layer transplantation analysis referenced in §5, which tests whether early-layer demographic encodings can produce causal effects when injected at late layers (and vice versa). The analysis controls for the depth-proximity confound: late-layer perturbations are constructed in a higher-norm region of the residual stream than early-layer ones, so apparent late-layer dominance could in principle reflect magnitude rather than location. Norm-matching removes this confound by rescaling each perturbation vector to match the typical norm of vectors natively constructed at the injection layer.

**Setup.**  For each contrastive pair ($N = 400$ per layer-pair), we constructed the top-50 SAE feature perturbation vector at one layer (the vector layer) and injected it at another (the injection layer) as a residual-stream addition at the final token position. We tested three early-layer vector layers ($L_9$, $L_{14}$, $L_{18}$) against late-layer $L_{36}$ in both directions, yielding six layer-pair combinations. Each combination was run in two conditions: raw (the natively-constructed vector) and norm-matched (rescaled per-pair to match the median norm of vectors natively constructed at the injection layer). Each condition has its own Random-B baseline (top-K $|\delta|$ applied to randomly selected active features at the vector layer), injected at the same injection layer.

**Residual-stream norm growth.**  Table **??** reports the residual-stream norm distribution at the final token position across analyzed layers, computed across $N = 100$ validation prompts. Median norm grows from 80 at $L_5$ to 611 at $L_{36}$, a 7.6× increase; the $L_9 \to L_{36}$ growth ratio specifically is 5.5×. This depth-dependent norm scaling motivates norm-matching as a control.

-stream norm by layer (final token position, $N = 100$ prompts).

| Layer | Depth | Median | Mean | SD | P10 | P90 |
|-------|-------|--------|------|-----|-----|-----|
| $L_5$ | 12% | 80.0 | 80.0 | 0.4 | 79.4 | 80.7 |
| $L_9$ | 22% | 111.4 | 111.5 | 1.1 | 110.2 | 112.8 |
| $L_{14}$ | 34% | 167.5 | 167.2 | 2.0 | 164.5 | 169.8 |
| $L_{18}$ | 44% | 237.5 | 238.4 | 4.2 | 233.3 | 243.6 |
| $L_{20}$ | 49% | 283.3 | 283.4 | 4.8 | 277.2 | 290.4 |
| $L_{27}$ | 66% | 391.9 | 390.6 | 12.4 | 371.1 | 404.2 |
| $L_{32}$ | 78% | 486.8 | 485.1 | 18.0 | 461.5 | 507.3 |
| $L_{36}$ | 88% | 610.9 | 607.5 | 19.1 | 583.5 | 629.6 |

**Per-layer-pair results.**  Table **??** reports specificity (demographic recovery minus Random-B recovery) for each layer-pair combination, raw and norm-matched. The pattern is asymmetric. Outbound (early $\to L_{36}$): all three early layers produce specificity indistinguishable from zero under both raw and norm-matched conditions. Inbound ($L_{36} \to$ early): raw specificity is large (15.5–34.4pp), but collapses substantially under norm-matching (0.0–6.1pp), with the largest residual signal at $L_{36} \to L_{18}$ (6.1pp, $p < 10^{-15}$).

-layer specificity per layer-pair ($N = 400$ per pair). Specificity is demographic minus Random-B recovery, expressed in percentage points (pp). NM = norm-matched.

| Direction | Layer pair | Raw spec | Raw $p$ | NM spec | NM $p$ |
|-----------|-----------|----------|---------|---------|--------|
| Outbound | $L_9 \to L_{36}$ | −0.2 | 0.18 | 1.8 | 0.05 |
| Outbound | $L_{14} \to L_{36}$ | 0.0 | 0.99 | −0.3 | 0.75 |
| Outbound | $L_{18} \to L_{36}$ | 0.0 | 0.92 | −0.1 | 0.92 |
| Inbound | $L_{36} \to L_9$ | 15.5 | $< 10^{-9}$ | 0.0 | 0.95 |
| Inbound | $L_{36} \to L_{14}$ | 21.8 | $< 10^{-10}$ | 1.4 | $< 10^{-4}$ |
| Inbound | $L_{36} \to L_{18}$ | 34.4 | $< 10^{-22}$ | 6.1 | $< 10^{-15}$ |

**Pooled results.**  Pooled across the three early layers ($N = 1200$): outbound raw specificity is −0.0pp ($p = 0.68$), outbound norm-matched is 0.5pp ($p = 0.38$, $d_z = 0.03$); inbound raw is 23.9pp ($p < 10^{-38}$,

$d_z = 0.39$), inbound norm-matched is 2.5pp ($p < 10^{-17}$, $d_z = 0.25$). The pooled outbound result is indistinguishable from zero in both conditions; the pooled inbound result drops from large to small under norm-matching but remains statistically detectable.

**Per-demographic breakdown (norm-matched, pooled).** Table 15 reports per-demographic specificity for outbound and inbound directions, norm-matched and pooled across the three early-layer pairings. The inbound residual is consistent across all five demographics ($p < 0.05$ in each case), with the largest effect for age (3.7pp) and the smallest for education (1.0pp). Outbound specificity is near zero in all demographics and reaches nominal significance in three cells (age, education) with mixed signs in others, consistent with sampling noise around zero rather than a coherent effect.

Table 15: P

er-demographic norm-matched specificity, pooled across the three early-layer pairings.

| Demographic | $N$ | Outbound (early $\to L_{36}$) | | | Inbound ($L_{36} \to$ early) | | |
|---|---|---|---|---|---|---|---|
| | | Demo | RB | Spec | Demo | RB | Spec |
| Income | 318 | $-1.2\%$ | $0.9\%$ | $-2.1$pp | $2.8\%$ | $-0.4\%$ | $3.2$pp |
| Age | 156 | $2.0\%$ | $-2.5\%$ | $4.5$pp | $4.1\%$ | $0.4\%$ | $3.7$pp |
| Gender | 207 | $-0.1\%$ | $-1.3\%$ | $1.2$pp | $2.0\%$ | $-0.5\%$ | $2.5$pp |
| Education | 258 | $1.1\%$ | $-1.0\%$ | $2.2$pp | $1.4\%$ | $0.4\%$ | $1.0$pp |
| Vote | 261 | $-2.6\%$ | $-1.6\%$ | $-1.0$pp | $2.0\%$ | $-0.5\%$ | $2.4$pp |

**Interpretation.** The asymmetry between directions, a clean outbound null and a small but non-zero inbound residual, supports the framing of encoding–causal independence between depths rather than the stronger claim that encoding and causal influence are governed by completely distinct mechanisms. Late-layer machinery cannot read early-layer demographic encodings in any direction we can detect; early-layer machinery can extract a small fraction of the demographic signal carried by late-layer vectors. We speculate this asymmetry reflects the polysemantic composite-axis structure of late-layer features documented in Appendix D.5: late-layer vectors carry correlated multi-demographic content with directions that early-layer-onward computation can partially use, while single-attribute early-layer encodings have no analogous structure to project onto late-layer machinery. We do not test this interpretation directly and flag it as speculative.

## F Bootstrapped orthogonalization Analysis

This appendix details the orthogonalization analysis referenced in §5.6, which tests whether per-pair top-K patching transplants correlated demographic information (bundling) rather than demographic-specific signal.

**Setup.** For each contrastive pair, we compute the orthogonalised perturbation vector $v_\perp = v - \sum_{j \in \text{off-target}} \langle v, \hat{d}_j \rangle \hat{d}_j$, where $\hat{d}_j$ is the unit-normalized encoding direction for off-target demographic $j$, computed from per-feature Cohen's $d_z$ thresholded at $|d_z| \geq 0.3$ on a disjoint held-out pool of 30 pairs per demographic. We re-run patching with both $v_\perp$ and norm-matched $v_\perp^{nm}$ (rescaled to $\|v\|$, controlling for projection-induced norm reduction). Disjoint working and held-out pools at the pair_key level avoid overlap-induced bias from the per-pair top-K selection.

**Bootstrap.** To bound magnitude estimates, we resample the held-out pool 10 times (30 pairs per demographic per resample) and report bootstrap means with 95% bootstrap confidence intervals.

**Pooled results.** In Gemma 2 9B at L36 ($N = 362$ pairs; 1,191 features in pool), original $v$ recovers 82.8% of the behavioral effect; bootstrap-mean $v_\perp^{nm}$ recovers 79.6% [95% CI: 77.7, 83.1]. Off-target loadings contribute a 3.2pp residual after norm matching [95% CI: $-0.3$, 5.1; SD 1.7pp]; 8 of 10 bootstraps yield $p < 10^{-5}$ on paired tests against $v$. The on-target component carries approximately 96% of the causal effect.

In Qwen 2.5 7B at L27 ($N = 301$ pairs; 290 features in pool), original $v$ recovers 98.8%; bootstrap-mean $v_\perp^{nm}$ recovers 84.7% [95% CI: 67.9, 91.8]. Off-target loadings contribute a 14.1pp residual [95% CI: 7.0, 30.9; SD 8.0pp]; 9 of 10 bootstraps yield $p < 0.05$. The on-target component carries approximately 85% of the causal effect.

**Per-demographic results.** Table 16 reports norm-matched residual contributions by demographic, with 95% bootstrap CIs. In both architectures, gender, age, and vote show 95% lower bounds above zero; education and income have CIs that cross zero.

Table 16: P

er-demographic norm-matched bundling residual (pp), bootstrap mean [95% CI].

| Demographic | Gemma 2 9B (L36) | Qwen 2.5 7B (L27) |
|---|---|---|
| Gender | 5.2 [0.8, 10.0] | 18.1 [3.2, 37.0] |
| Age | 5.1 [0.7, 9.8] | 15.9 [6.2, 30.4] |
| Vote | 3.1 [0.2, 6.0] | 21.5 [11.3, 39.5] |
| Education | 2.3 [−1.7, 7.0] | 7.3 [−2.8, 13.4] |
| Income | 2.1 [−1.3, 4.3] | 8.7 [−13.7, 31.5] |

**Encoding-direction cosine structure.** The geometric arrangement of demographic encoding directions differs across architectures (Table 17). In Gemma, the strongest alignment is education–vote (0.43 [0.34, 0.54]), with gender moderately anti-aligned to age, education, and vote (cosines −0.23 to −0.26). In Qwen, the strongest alignments form an income–age–education cluster (cosines 0.30–0.33), with gender anti-aligned to age (−0.37 [−0.58, −0.16]). Both architectures show late-layer composite-axis structure but in qualitatively different geometric arrangements, suggesting the polysemantic bundling pattern is not architecture-invariant.

Table 17: P

airwise encoding-direction cosines, bootstrap mean [95% CI]. Subset shown for illustration; full matrix in

| Pair | Gemma 2 9B (L36) | Qwen 2.5 7B (L27) |
|---|---|---|
| Education–Vote | 0.43 [0.34, 0.54] | 0.17 [−0.07, 0.34] |
| Income–Education | 0.28 [0.18, 0.38] | 0.30 [0.03, 0.50] |
| Income–Age | −0.11 [−0.19, −0.03] | 0.33 [0.21, 0.47] |
| Age–Education | −0.15 [−0.24, −0.03] | 0.33 [0.16, 0.48] |
| Age–Gender | −0.26 [−0.33, −0.17] | −0.37 [−0.58, −0.16] |
| Gender–Education | −0.23 [−0.36, −0.13] | −0.04 [−0.29, 0.18] |

supplementary materials.

**Interpretation.** The bundling effect direction is robust across both architectures and all bootstrap resamples; magnitude is precisely estimated in Gemma (±2pp at 95% CI) and uncertain in Qwen (±15pp at 95% CI), reflecting smaller encoding-direction sample sizes in Qwen. The Qwen NPE overshoot in §5 (132.9% at L27) is therefore partly explained by bundling against correlated demographic structure (14.1pp pooled) but not fully: additional contributions come from Qwen's higher general perturbation sensitivity at L27, Random-B baseline 43.0%) and the magnitude scaling inherent to per-pair top-K selection.

