# OpenReview forum: "Encoding Without Influence: Dissociating Demographic Representation from Causal Effect in Large Language Models"
_TMLR — Accepted by TMLR_

### Review · Reviewer_3QWG · 2026-04-27

**Summary Of Contributions:**

This work studies how LLMs encode demographic information and drive variation in model outputs. Using contrastive prompt pairs where one contains a specific attribute while the other does not, the authors (1) study representation via sparse autoencoders, and (2) analyze LLM output changes through intervention methods. Specifically, they evaluate the causal role of identified features mechanistically using patching, steering, and ablation. The main finding is the encoding–causal dissociation: early layers encode the demographic information, while later layers control the behavioral shifts despite weaker demographic encoding.

**Audience:**

Yes

**Audience Explanation:**

1. Findings are relevant to model interpretability
2. Provides useful implications for bias auditing and related applications

**Claims And Evidence:**

Yes

**Claims Explanation:**

1. Presents a clear experiment setup and result breakdown
2. Experiments use multiple causal interventions (patching, steering, and ablation) to study different aspects of how internal features influence model behavior

**Requested Changes:**

**Weaknesses / Questions**

1. Prior work has also established closely related claims, e.g., probing doesn’t imply casual influence [1, 2], and casual tracing localizes factual associations in mid-layer MLP [3]. In this sense, the contribution of this work is narrower.
2. Experiments are conducted on numeric survey responses. It is unclear how the findings transfer to open-ended generation, where model outputs may also be conditioned by chain-of-thought prefixes or other intermediate context. Are late-layer demographic features still causally active in open-ended generation?
3. The handling of edge-case token “1” between the answers 1 and 10 (Sec 4.2) may introduce substantial noise, especially since 1 and 10 are opposite ends of the response scale. Sequence-level scoring would be more principled.
4. How would findings change if more subtle / shorter / less extreme demographic descriptions, are used?

---

*References*

[1] Probing the Probing Paradigm: Does Probing Accuracy Entail Task Relevance?\
[2] Amnesic Probing: Behavioral Explanation with Amnesic Counterfactuals\
[3] Locating and Editing Factual Associations in GPT

---

> ### Author Response · Authors · 2026-05-18
> **Response to Reviewer 3QWG**
>
> We thank the reviewer for the careful engagement; we respond to each point below.
>
> **Point 1: Prior work overlap**
>
> We acknowledge the antecedents the reviewer identifies. The probing-vs-causal lineage (Ravichander et al., 2021; Elazar et al., 2021) and causal localisation of factual associations (Meng et al., 2023) are foundational for our methodology. Our specific contribution is positioned narrowly relative to these: the dissociation (a) holds across multiple detectability measures rather than being specific to probes; (b) concerns demographic influence on normative outputs rather than factual recall; (c) maps a depth profile (encoding peaks early, causal mediation peaks late) that prior work has not characterised; and (d) documents polysemantic composite-axis structure at the causally-active layers (§5.6, Appendix F).
>
> We have revised §2.2 to make these distinctions explicit, separating the domain difference (normative judgment vs factual recall) from the methodological extensions (multiple detectability measures, depth profile, feature-level structure at causally-active layers).
>
> **Point 2: Generalisation to open-ended generation**
>
> We do not claim our findings transfer to open-ended generation and have added a note to §6.3 acknowledging this scope restriction. Numeric survey responses were chosen for two reasons: they provide a clean causal target (a distribution over ordinal values, directly measurable via expected value), and the contrastive paired structure is statistically tractable in a way that open-ended generation is not, where defining an equivalent dependent variable requires additional design choices (sentiment score, toxicity classifier, semantic similarity, etc.). Whether late-layer demographic features remain causally active under chain-of-thought or open-ended decoding is a natural extension we flag as future work.
>
> **Point 3: Token "1" disambiguation and sequence-level scoring**
>
> We agree that sequence-level scoring is more principled than the 50/50 heuristic and ran a sensitivity analysis on 240 stratified pairs (12 per demographic × domain cell, scale-11 questions only). For each prompt we computed EV under both methods: single-token probabilities for values 0, 2–9; for values 1 and 10, joint probabilities P(token_1 | prompt) × P(end_of_response | prompt+"1") and P(token_1 | prompt) × P(token_0 | prompt+"1") respectively. For patched conditions the second-token distribution was computed under the same residual-stream patch as the first token.
>
> The two methods agree closely on per-prompt EV (Pearson r = 0.97, N=480 prompts), on contrastive ΔEV (r = 0.97, 95.8% sign agreement), and on pooled patching recovery (original 67.7%, sequence-level 65.8%, mean shift −1.9pp [95% bootstrap CI: −6.1, +1.3] on N = 192 pairs above the effect threshold). The pooled CI crosses zero and four of five demographics individually show CIs spanning zero. The encoding–causal dissociation finding is robust to scoring-method choice. The one exception is vote × values-domain questions specifically (N=10 pairs, −32pp shift), where answer distributions concentrate at scale boundaries and the 1/10 disambiguation has greatest leverage; the vote-only pooled shift is −7.7pp [−18.4, −0.3], and other vote domains show shifts ≤ 3.4pp.
>
> We have added a paragraph to §4.2 describing both methods, flagged vote × values magnitude as scoring-method-sensitive in §6.3, and included the full sensitivity analysis in Appendix D.2.
>
> **Point 4: Generalisation to subtler/less extreme demographic descriptions**
>
> §3 acknowledges this scope restriction directly: extreme contrasts were chosen by design to maximise activation differences for feature discovery, not to estimate realistic effect sizes. This follows standard practice in causal tracing, where unambiguous stimuli are used to identify mediating components (Meng et al., 2023). Whether the identified features generalise to subtler demographic cues is genuinely open and we have flagged it in §6.3 as a scope restriction alongside non-Western contexts and open-ended generation.
>
> One existing piece of evidence is partial: §3 reports paraphrase-robustness testing on a held-out subsample (30 respondents × 20 questions, mean ρ = 0.79 across wordings), which confirms identified features track demographic content rather than specific phrasing. This addresses surface-form variation but not contrast intensity —whether features identified under PhD-vs-no-secondary education still mediate effects under bachelor's-vs-high-school is the natural next test and one we intend to run as a follow-up.

---

### Review · Reviewer_BxgT · 2026-05-04

**Summary Of Contributions:**

The paper investigates whether the layers that most strongly *encode* demographic information in instruction-tuned LLMs are the same layers at which demographic information *causally shapes* outputs on normative survey questions.

Using SAEs and three causal interventions (activation patching, feature steering, targeted ablation), the authors trace this in Gemma, Qwen, and Llama models with prompts covering five policy domains.

The authors show that encoding and causal effect dissociate: early and middle layers show the strongest demographic encoding, yet interventions there produce no measurable effect on outputs above variance-matched controls; conversely, late-layer features with weaker encoding largely recover the behavioral demographic effect via patching. The authors test different encoding profiles across architectures. Across settings, causal influence concentrates in roughly the final third of network depth.

Other findings are:
- The demographic ranking and recovery magnitude shift across the five policy domains.
- Late-layer causal features form polysemantic composite social axes, with opposite-sign loadings across correlated demographics (e.g., younger/wealthier/educated co-loading), arguing that single-attribute probes would miss them.

**Audience:**

Yes

**Audience Explanation:**

The paper has multiple interesting findings:

- main story, the gap between representation and causal effect
- domain modulation of demographic influence
	- implication that aggregate fairness audits can pass while domain-specific audits fail is actionable and well-supported by the data shown
- late-layer features as correlated social bundles
	- highly relevant to concept erasure: e.g., if "the gender direction" at late layers is in fact a younger/wealthier/educated/female composite, then removing it removes more than gender, and standard single-attribute erasure methods are quietly modifying correlated attributes.

each on its own justifies interest

**Broader Impact Concerns:**

authors include a good broader impact statement

**Claims And Evidence:**

No

**Claims Explanation:**

overall careful, well-controlled empirical paper

key claims and evidence:
- late-layer interventions reliably modulate demographic-conditioned outputs across three architectures, with clean controls.
- mechanistic interpretation (that encoding and causal deployment are governed by _distinct mechanisms_) is suggestive but not isolated from the depth–proximity confound (W2).

### strengths

- S1. contrastive prompting data design is clean
	- matched persona pairs with a single varying attribute, real ESS values for non-target demographics, and paraphrase-robustness checks make the behavioral signal interpretable
- S2. careful controls
	- random-A, random-B, cross-condition, and variance-matched null each address a distinct alternative. reporting specificity gaps is good.
	- variance-matched null baseline is the right control for the "any high-variance perturbation would do this" alternative
- S3. tested across three architectures
	- different SAE families (PT Gemma Scope, IT SAELens for Qwen, additional Llama replication) and different alignment procedures. The L20 IT-variant comparison in Gemma is a useful internal control on PT–IT mismatch.
- S4. late-layer causal features as polysemantic composite social axes (Appendix D.5)
	- direct relevance to concept-erasure work: if "the gender direction" at late layers is a younger/wealthier/educated/female composite, then standard single-attribute erasure methods are quietly modifying correlated attributes.
	- deserves more prominence than it currently gets (substantive finding in its own right)

### weaknesses

- W1: (minor) critique applies more directly to the concept-erasure / probe-direction literature than to deployed fairness audits:
	- in standard practice, fairness conclusions are not typically based on encoding strength, but on either behavioral auditing and final hidden state or specific MLPs chosen by ablation studies, not by probe accuracy
	- but it's relevant to papers like "we found the gender direction, here's how to remove it" (concept-erasure and direction-finding)

- W2: depth–proximity confound is not neutralized
	- late-layer interventions have fewer downstream blocks to attenuate them; the paper's argument against pure proximity (§5.2) (early-layer encoding is "strong enough" that some leakage should appear) is plausible but not a controlled comparison
		- the authors argue something like this: if early layers really encoded the same demographic information that late layers use, you'd see at least a faint causal echo proportional to encoding strength, attenuated by depth. You don't see that. So early-layer encoding must be a different thing.
			- the argument assumes (a) downstream attenuation acts roughly multiplicatively on perturbation magnitude, and (b) encoding strength translates monotonically into perturbation magnitude
			- however, the residual stream is linear between blocks but not through them:  nonlinear units (LayerNorm, softmax, MLP gating) can act as thresholds
	- alternative hypothesis: early-layer features participate in producing outputs, but downstream computation refines, smooths, or replaces the perturbation before readout, so the intervention does not survive to the logits
		- cleanest test would be cross-layer perturbation transplantation: take a late-layer perturbation and inject it at an early layer
			- pure proximity predicts the late-layer-effective vector also fails when injected early; representational reorganization predicts it might survive
			- residual-stream linearity makes this technically straightforward and it is the natural experiment for the central claim

- W3. polysemanticity inflates NPE in ways the controls don't catch.
	-  in Appendix D.5 the authors find polysemanticity: late-layer causal features as composite social axes (younger/wealthier/educated/female loading together with opposite signs across demographics)
	- per-pair top-K selection then transplants the bundle, not the on-target demographic component.
		- Qwen's NPE and gender overshoots could be explained by off-target bundling: the contrastive prompt holds non-target demographics nominally constant, but the model has reason to re-correlate them via the composite axis?
	- Random-B controls for perturbation magnitude but not for off-target demographic loading. Orthogonalizing v against off-target encoding directions (using the per-feature loadings already computed in encoding matrix in D.5) and re-running patching would adjudicate "demographic-specific mediation" versus "joint-social-signal mediation."
		- then two outcomes are possible: 1) $\text{NPE}(v_{\perp}) ≈ \text{NPE}(v)$ (off-target loadings are noise) or 2) much smaller, both would be interesting, but support different bias-auditing implications.


- W4. Llama ablation null undermines "three-way convergence across architectures."
	- redundant-encoding hypothesis offered in §5.6 is reasonable but untested
	- cross-architecture convergence is presented as more uniform than the data support;
		- a direct sequential-ablation test (ablate top-K, then ablate the next disjoint top-K, check whether effect accumulates) would test the redundancy claim

phenomenology meets TMLR's evidence bar; the strong mechanistic framing currently outruns it.

The gap is closeable on existing data (without new experiments) with orthogonalization (W3), cross-layer perturbation transplantation (W2), or sequential ablation in Llama (W4)

**Requested Changes:**

### critical

- C1. depth-proximity confound (W2).
	- inject the late-layer-effective $v_{\text{late}}$ at an early layer (e.g., L36 → L9 in Gemma). pure proximity predicts failure; representational reorganization predicts partial survival. Reuses existing perturbation vectors.
	- *or* drop the claim of "distinct mechanisms" and "representationally reorganised between encoding and causal deployment" (§5.2, §6.1). replace with the descriptive claim the data support directly: "the layers most detectable by encoding strength are not the layers at which intervention modulates outputs."

- C2. polysemantic bundling in NPE (W3).
	- project $v$ onto the orthogonal complement of off-target encoding directions from the D.5 matrix, $$v_\perp = v - \sum_{j \in \text{off-target}} \langle v, \hat{d}_j\rangle \hat{d}_j,$$ and re-run patching.
	- *or* in §6.2, qualify "demographic-specific causal mediation" to "mediation by features that load on the target demographic alongside correlated social attributes." Frame the auditing implication accordingly: late-layer features mediate _bundles_ of demographic information, and whether removing the on-target component alone would reproduce the effect is not established. Note Qwen's NPE > 100% as plausibly bundling-driven.

- C3. Llama ablation null (W4).
	- sequential ablation: ablate top-K, then the next disjoint top-K, check whether reduction accumulates. Tests the redundancy hypothesis offered in §5.6.
	- *or* drop* "three-way convergence across architectures" framing (Contribution 2, §5.2, Conclusion).

### improving

- reframe the critiqued pipeline (§1, §6.2) as targeting the concept-erasure / probe-direction literature (Ravfogel 2020, Belrose 2023 LEACE) rather than deployed fairness audits. Makes the contribution sharper.
- promote D.5 (composite social axes) to a §5 subsection with Figure 12 in the main paper. Strong finding for concept-erasure work and somewhat under-marketed.

### minor / editorial

- §4.1: stray space in "(pretrained–instruction-tuned )mismatch"; inline expansion of "MLP (Multi-layer perceptron)" reads awkwardly mid-sentence.
- Table 8 vs. §5.2 Qwen $|d_{\text{enc}}|$ values: clarification ("within-group top-50 vs. layer-wide mean") arrives after the discrepancy. Move adjacent to first mention.
- §5.5: Qwen's elevated Random-B is presented as evidence for polysemanticity, but could equally reflect higher perturbation sensitivity (which §5.5 itself notes for L7). Make the inference explicit or qualify.
- Bouchaud & Ramaciotti 2025: clarify in related work whether their causal claim was layer-stratified, since this paper's contribution depends on depth-stratification being new.
- §6.3 (Limitations) could absorb the depth-proximity, polysemanticity-inflation, and Llama-redundancy caveats if the corresponding experiments are not run.

---

> ### Comment · Reviewer_BxgT · 2026-05-18
> **revision clears remaining issues, good paper!**
>
> The revision addresses all three critical concerns. C1 is handled by the cross-layer transplantation in Appendix E; the norm-matching control is essential and correctly implemented, and the asymmetric result supports the encoding–causal independence framing without overclaiming. The minor residual in §6.1 ("functionally distinct rather than transformations") slightly outruns the data but is not worth blocking on. C2 is handled by the bootstrapped orthogonalization in Appendix F; Gemma's estimate is tight, Qwen's is wide but the authors acknowledge this, and the §6.2 implications are now appropriately qualified. C3 is resolved by reframing rather than new experiments. Promoting §5.6 to the main paper was the right call.

---

> ### Author Response · Authors · 2026-05-18
> **Response to Reviewer BxgT**
>
> We thank the reviewer for the detailed methodological concerns, which we have taken seriously. The critical assessment that our claims were not sufficiently supported by evidence led us to perform two new experiments on existing data (cross-layer transplantation for C1 and bootstrapped orthogonalisation for C2) and to substantially revise the cross-architecture framing (C3).
>
> 1. **C1 (depth-proximity confound):** direct experimental control via cross-layer transplantation with norm-matched perturbations. Demographic features from early layers, when norm-matched and injected at L36, produce zero causal specificity (pooled 0.5pp, p = 0.38, N = 1200 pairs). The reverse direction (L36 → early) shows a small but non-zero norm-matched residual (2.5pp, dz = 0.25), which we interpret as evidence that early and late demographic encodings are not functionally interchangeable rather than as evidence for completely distinct mechanisms. This rules out depth-proximity as a sufficient explanation for the encoding-side null at early layers. We report this analysis for Gemma 2 9B; the analogous test for Qwen was not run for this revision.
>
> 2. **C2 (polysemantic bundling):** direct experimental decomposition of on-target versus correlated-demographic causal contribution via bootstrapped orthogonalisation. After removing off-target demographic directions, on-target features still recover 96% (Gemma) and 85% (Qwen) of the behavioural effect, bounding the bundling contribution at ~3pp and ~14pp respectively. The bundling account is therefore real but partial; demographic-specific mediation remains the dominant component in both architectures.
>
> 3. **C3 (cross-architecture generalisation):** softened to "two architectures with partial replication on a third." Llama replicates the encoding side of the dissociation but not the ablation side; we no longer claim three-way convergence on causal mediation.
> The qualitative pattern of encoding–causal dissociation now rests on two empirical controls that did not exist in the original submission, and the cross-architecture claim has been calibrated to what the data support. We hope this addresses the reviewer's central concern.

---

> > ### Author Response · Authors · 2026-05-18
> > **In-depth response to Reviewer BxgT**
> >
> > **C1. Depth–proximity confound (cross-layer transplantation)**
> >
> > Appendix E reports the full analysis. We constructed top-50 SAE feature perturbation vectors at three early layers (L9, L14, L18) and at L36, and tested all six layer-pair combinations in both directions, raw and norm-matched. Norm-matching is essential: residual-stream norms grow ~5.5× from L9 to L36 (Table 15), so any apparent late-layer dominance under raw injection could reflect magnitude rather than location.
> >
> > The pattern is asymmetric. Outbound (early → L36): pooled raw specificity −0.0pp (p = 0.68); norm-matched 0.5pp (p = 0.38, dz = 0.03) early-layer encodings transplanted to L36 produce no detectable causal effect. Inbound (L36 → early): raw 23.9pp collapses to 2.5pp under norm-matching (p < 10⁻¹⁷, dz = 0.25)  the bulk is magnitude-driven, but a small residual remains.
> > We have recalibrated the §5.2 framing accordingly: the asymmetry supports that early and late demographic encodings are "not functionally interchangeable" and "not transformations of one another," rather than the stronger claim of completely distinct mechanisms. Appendix E's interpretation paragraph flags the speculation about composite-axis structure underlying the small inbound residual as untested.
> >
> > **C2. Polysemantic bundling in NPE (orthogonalisation)**
> >
> > Appendix F reports the bootstrapped orthogonalisation analysis. For each contrastive pair, we compute v_⊥ = v − Σ_{j ∈ off-target} ⟨v, d̂_j⟩ d̂_j, where d̂_j is the unit-normalised encoding direction for off-target demographic j (estimated from a disjoint held-out pool of 30 pairs per demographic, |dz| ≥ 0.3). We re-run patching with norm-matched v_⊥^nm (rescaled to ‖v‖) and bootstrap the held-out pool 10 times for 95% CIs.
> > The direction is robust; the magnitude is architecture-dependent. In Gemma at L36 (N = 362), off-target loadings contribute 3.2pp [95% CI: −0.3, 5.1]  the CI crosses zero and demographic-specific mediation survives essentially intact. In Qwen at L27 (N = 301), off-target loadings contribute 14.1pp [95% CI: 7.0, 30.9]  non-trivial but uncertain by approximately ±15pp, reflecting smaller encoding-direction sample sizes.
> >
> > We have qualified the §6.2 framing in line with the reviewer's proposed wording: late-layer features mediate "bundles of demographic information," and whether removing the on-target component alone would reproduce the effect is established for Gemma but only partially so for Qwen. §5.5 now presents both candidate accounts of the Qwen NPE > 100% overshoot (bundling and higher general perturbation sensitivity) and notes that C2 supports the bundling account without ruling out the sensitivity account.
> > The per-demographic breakdown (Table 18) tracks the §5.6 feature-level structure: gender, age, and vote show 95% lower bounds above zero in both architectures, matching the demographics with the strongest opposite-direction encoding in Figure 6.
> >
> > **C3. Llama ablation null and three-way convergence framing**
> >
> > We have softened the cross-architecture framing throughout the manuscript rather than running the sequential-ablation test, which would require additional experiments beyond the scope of this revision:
> > 1. Contribution 2 (§2.3) now reads "validated against matched controls in Gemma and Qwen, with partial replication on Llama (§5.7)."
> > 2. §5.2 separates Llama explicitly: "The convergence across two architectures provides strong evidence... Llama's pattern (strong encoding, weaker ablation) is discussed separately in §5.7."
> > 3. §5.7 presents the distributed-encoding hypothesis as a candidate explanation with a directly testable prediction (sequential ablation of disjoint top-K batches should produce accumulating reductions), flagged as future work rather than offered as the explanation.
> > 4. §6.3 lists the non-significant ablation as a limitation alongside the redundant-encoding hypothesis.
> > We agree the sequential-ablation test is the right next step and intend to run it as part of follow-up work.

---

> > > ### Author Response · Authors · 2026-05-18
> > > **In-depth response to Reviewer BxgT**
> > >
> > > **Improving suggestion: Reframing toward concept-erasure literature**
> > > We agree the concept-erasure framing is sharper. Two implications of our findings bear directly on this literature: projection-based erasure at layers with strong encoding but weak causal mediation (early-to-middle layers in our models) may remove detectable demographic information without removing causal influence; and at late layers where causal mediation concentrates, the polysemantic bundling structure means single-concept erasure operates on cross-covariance subspaces that necessarily span correlated demographic attributes, so single-attribute erasure involves trade-offs rather than clean isolation.
> > > We have engaged with this literature at a targeted level: a new §2.2 paragraph situates our findings within the concept-erasure programme, citing INLP (Ravfogel et al. 2020), LEACE (Belrose et al.), and the broader probing/causal-tracing lineage (Ravichander et al. 2021; Elazar et al. 2021; Meng et al. 2023); §2.3 Contribution 5 notes downstream implications for literatures assuming single-direction concept representations; a new §6.2 paragraph draws out the two concrete implications above; and the encoding-presence / causal-mediation decomposition introduced for Reviewer aEwm (§1, §6.1) aligns our terminology with concept-erasure terminology more precisely. A full reframing of the paper's primary thesis is beyond this revision but is a direction we intend to develop.
> > >
> > > **Improving suggestion: Promoting D.5 (composite social axes) to the main paper**
> > >
> > > We agree this finding was under-marketed, particularly in light of the C2 orthogonalisation results: the polysemantic composite-axis structure has direct causal consequences (approximately 14pp bundling in Qwen, 3pp in Gemma, with gender contributing robustly in both architectures). The D.5 analysis has been promoted to a new main-paper subsection §5.6 (Composite Social Axes at Causal Layers), with the Gemma L36 encoding heatmap now in the main paper as Figure 6 and explicit cross-references to §5.5 and §6.2. Appendix D.5 is retained for the per-architecture breakdowns including the Qwen encoding matrix and response-position concentration analysis.
> > >
> > > **Editorial corrections and minor clarifications**
> > > 1. The stray space in "(pretrained–instruction-tuned )mismatch" in §4.1 is fixed (now reads "pretrained–instruction-tuned (PT-IT) mismatch"); the awkward inline "MLP (Multi-layer perceptron)" expansion has been removed.
> > > The Table 8 / §5.2 metric clarification Table 8 reports within-group top-50 mean |d_enc| while §5.2 references the layer-wide mean  is now placed adjacent to the first mention in §5.2.
> > >
> > > 2. §5.5 now presents both candidate accounts of the Qwen NPE >100% overshoot: (i) polysemantic composite-axis structure carrying multi-demographic content, supported by the C2 orthogonalisation results, and (ii) higher Qwen-specific perturbation sensitivity that would inflate Random-B regardless; C2 supports (i) without ruling out (ii). The L7 elevated Random-B is flagged as independent evidence for (ii).
> > >
> > > 3. On Bouchaud & Ramaciotti (2025): §2.1 now clarifies that their causal validation is performed at a single layer per model  the probing-optimal layer establishing that linear demographic encodings are causally active at that layer but not characterising the depth distribution of encoding strength vs. causal influence. Our work complements theirs by examining this depth distribution directly.
> > >
> > > 4. §6.3 (Limitations): C1 and C2 are addressed empirically rather than absorbed into limitations. The Llama distributed-encoding hypothesis is now an explicit limitation, alongside a note that the Qwen polysemantic bundling magnitude carries methodological uncertainty although the directional conclusion is robust across bootstraps.

---

### Review · Reviewer_aEwm · 2026-05-05

**Summary Of Contributions:**

The authors present study how demographic information is represented in large language models, specifically how the representations of demographic information effects the output of the models in contexts where normative judgement is applied. The main findings are that representational strength of a demographic feature does not coincide (in terms of location in the model's layers) with this feature's causal influence on the models final output. This dissociation between the demographic representation features and those features exerting a causal effect on outputs is prevalent across three studied models. As a consequence, the authors argue that representational detection is insufficient for bias auditing, which must rather be validated according to their proposed causal effect strategy and be considered on a per-domain basis.

*Disclaimer*: I am not an expert in LLM representation studies and can thus not judge the contribution of the methodology from the lens of that field. My review focusses on general methodological aspects as well as the interpretation of the presented results.

## Strengths
- The paper is well written, central claims are clearly mapped out (and repeated at key sections of the paper) and conclusions as well as limitations of the presented work are presented without obfuscation.
- The methodology of the paper seems sound. The authors build upon (what to my judgement seems to be) existing approaches for studying LLM representations, provide a rigorous argument for their proposed approach (assumption of linearity of representations, interventions/perturbations to study causal effects, multiple baselines, statistical analysis of results) and discuss their findings candidly without making claims for which they do not provide evidence.
- The discussion of implications of their findings is a valuable contribution and directly translates the findings of this work into guidance for downstream applications.

## Weaknesses
- At time the papers is repetitive and could be condensed.
- Figure 3A: legend is placed awkwardly
- Certain concepts are missing references (Cohen's $d$, BH-correction)

**Audience:**

Yes

**Audience Explanation:**

LLMs are being deployed en-masse at the current moment and almost surely being used in contexts where demographic information informs their output for sensitive queries, making this a timely work. The references provided in the paper also show that this work is embedded in a larger research field, making its findings relevant for the researchers active there.

**Claims And Evidence:**

Yes

**Claims Explanation:**

The authors provide a rigorous evaluation of their claims surrounding representational strength and causal effect of demographic information, as well as an in-depth discussion of the results of their experiments. Claims are not over-stated and limitations are discussed.

**Requested Changes:**

Not critical, but I would be interested to know if the main message of the paper could actually be that representational strength is a misguided metric to begin with. My interpretation of such metrics are that they imply that representational strength is called *strength* to imply its effect on something, which your work clearly demonstrates is not the case. Could you discuss how to disentangle these concepts? Should we do away with the measures and just re-name your measure of causal effect the representational strength?

---

> ### Author Response · Authors · 2026-05-18
> **Response to Reviewer aEwm**
>
> We thank the reviewer for the careful engagement with the manuscript and for the discussion question on representational strength, which directly shaped one of our conceptual revisions.
>
> **Discussion question on "representational strength"**
>
> The reviewer's diagnosis  that "representational strength" was named "strength" because the implicit assumption was that strong encoding implies strong functional impact, and that our findings refute this implication  is, we believe, exactly correct.
>
> Our paper's central conceptual contribution is the decomposition of "representational strength" into two distinct quantities:
>
> (1) encoding presence :  whether demographic information is detectable in a representation, measurable by probe accuracy, SAE feature activation contrasts, Cohen's d, etc.
>
> (2) causal mediation : whether that information functionally drives model behaviour, measurable by patching recovery, ablation effects, and steering shifts.
>
> These have historically been conflated under "representational strength," and the conflation imports a behavioural claim the evidence does not support. Our finding  early layers show strong encoding presence but near-zero causal mediation, late layers show weaker encoding presence but strong causal mediation  directly refutes the implicit conflation.
> We have revised §1, §2.2, and §6.1 to make this decomposition explicit as a conceptual contribution. What bias auditing requires is causal mediation, not encoding presence; probe-style metrics implicitly assume equivalence between the two, and our findings show they are not equivalent.
>
> We have not gone as far as the reviewer's suggested rename (replacing "representational strength" with our causal measure) because the dual-quantity framing is more precise: encoding presence remains a meaningful property of representations, just one that does not by itself license behavioural claims. The decomposition is the contribution, not the elimination of one term.
>
> **Other revisions in response to this review:**
>
> We have also addressed the reviewer's other comments: added citations for Cohen's d and the Benjamini–Hochberg procedure; repositioned the Figure 3A legend; and made a compression pass across §1, §5, and §6 to reduce repetition of the core dissociation claim, retaining instances only where they serve distinct functions (announcement, evidence presentation, theoretical interpretation, summary).

---

### Author Response · Authors · 2026-05-18
**Summary of Revisions**

We thank all three reviewers for their careful and constructive engagement with the manuscript. The reviews converged on several actionable concerns, and we have made substantive revisions in response.

1. Two new analyses on existing data address Reviewer BxgT's central methodological concerns: cross-layer transplantation (new Appendix E) directly tests and rules out the depth-proximity confound as a sufficient explanation for the encoding–causal dissociation, and bootstrapped orthogonalisation (new Appendix F) quantifies the contribution of polysemantic bundling, showing that demographic-specific mediation remains the dominant component in both Gemma (96%) and Qwen (85%). The cross-architecture framing has been softened throughout to "two architectures with partial replication on a third," reflecting Llama's non-significant ablation result.

2. In response to Reviewer aEwm's discussion question on whether "representational strength" conflates two distinct quantities, we have introduced the encoding-presence / causal-mediation decomposition as an explicit conceptual contribution (§1, §6.1). Per Reviewer BxgT's improving suggestions, we have engaged with the concept-erasure literature at a targeted level (§2.2, §6.2) and promoted the composite social axes analysis from Appendix D.5 to a new main-paper subsection §5.6 with Figure 6 in the main text.

3. In response to Reviewer 3QWG, we have added a sequence-level scoring sensitivity analysis (Appendix D.2) confirming the encoding–causal dissociation is robust to scoring-method choice in aggregate, with one boundary-concentrated subset flagged as scoring-method-sensitive. We have also revised §2.2 to position our contribution more precisely relative to the probing-vs-causal and causal-tracing lineages.

**All changes in the revised manuscript are highlighted in blue. Detailed per-reviewer responses follow as replies to each review.**

---

### Decision · Action_Editor_8rk5 · 2026-06-05

**Recommendation:** Accept as is

**Audience:**

Yes

**Audience Explanation:**

This work extends prior works to a new domain and is timely. The community would beneifit from the new knowledge this work adds.

**Claims And Evidence:**

Yes

**Claims Explanation:**

All reviewers agree that after revision, the claims are well supported.